# In Vitro and In Vivo Antioxidant and Immune Stimulation Activity of Wheat Product Extracts

**DOI:** 10.3390/nu17020302

**Published:** 2025-01-16

**Authors:** Beatrice Mengoni, Federica Armeli, Emily Schifano, Sabrina Antonia Prencipe, Laura Pompa, Fabio Sciubba, Elisa Brasili, Ottavia Giampaoli, Francesco Mura, Massimo Reverberi, Marzia Beccaccioli, Alessandro Pinto, Maria De Giusti, Daniela Uccelletti, Rita Businaro, Giuliana Vinci

**Affiliations:** 1Department of Medico-Surgical Sciences and Biotechnologies, Sapienza University of Rome, 04100 Latina, Italy; beatrice.mengoni@uniroma1.it (B.M.); federica.armeli@uniroma1.it (F.A.); 2Department of Human Sciences, European University of Rome, 00163 Rome, Italy; 3Department of Biology and Biotechnologies “C. Darwin”, Sapienza University of Rome, 00185 Rome, Italy; emily.schifano@uniroma1.it (E.S.); laura.pompa@uniroma1.it (L.P.); daniela.uccelletti@uniroma1.it (D.U.); 4Department of Management, Sapienza University of Rome, 00161 Rome, Italy; sabrinaantonia.prencipe@uniroma1.it (S.A.P.); giuliana.vinci@uniroma1.it (G.V.); 5Department of Environmental Biology, Sapienza University of Rome, 00185 Rome, Italy; fabio.sciubba@uniroma1.it (F.S.); elisa.brasili@uniroma1.it (E.B.); f.mura@uniroma1.it (F.M.); massimo.reverberi@uniroma1.it (M.R.); marzia.beccaccioli@uniroma1.it (M.B.); 6NMR-Based Metabolomics Laboratory (NMLab), Sapienza University of Rome, 00185 Rome, Italy; 7Department of Chemistry, Sapienza University of Rome, 00185 Rome, Italy; ottavia.giampaoli@uniroma1.it; 8Department of Experimental Medicine, Sapienza University of Rome, 00161 Rome, Italy; alessandro.pinto@uniroma1.it; 9Department of Public Health and Infectious Diseases, Sapienza University of Rome, 00185 Rome, Italy; maria.degiusti@uniroma1.it

**Keywords:** wheat, microglia polarization, *C. elegans*, oxidative stress, inflammation, innate immune response, functional foods

## Abstract

Background/Objectives: Inflammation and oxidative stress are the main pathogenetic pathways involved in the development of several chronic degenerative diseases. Our study is aimed at assessing the antioxidant and anti-inflammatory activity of hydroalcoholic extracts obtained from wheat and its derivatives. Methods: The content of total phenolic and total flavonoid compounds and antioxidant activity were carried out by ABTS and DPPH assays. The ability of wheat extracts to promote microglia polarization towards an anti-inflammatory phenotype was evaluated analyzing the increased expression of anti-inflammatory markers by real-time qPCR and immunofluorescence assays. Antioxidant activity of all the extracts was evaluated in *C. elegans* by analyzing ROS levels and the expression of the antioxidant enzymes GST-4 and SOD-3 by real-time qPCR and fluorescence experiments. The expression of key genes involved in the innate immune response and stress resistance pathways—*daf-16*, *sek-1*, and *pmk-1*—was evaluated by real-time qPCR. Results: Wheat extracts showed the ability to polarize microglia cells towards an anti-inflammatory phenotype, even after the addition of LPS. An antioxidant response was detected both in microglia and in *Caenorhabditis elegans* nematode, where the extracts also implemented an anti-stress resilience response and stimulated the innate immunity. Conclusions: The present study shows that wheat seeds, flour, chaff, and pasta present anti-inflammatory as well as antioxidant activities and may be considered as prospective positive health agents for the preparation of functional foods. Moreover, the valorization of by-products from agricultural and agro-industrial activities would also have significant implications in terms of circular economy.

## 1. Introduction

Extensive literature has documented the advantageous effects of bioactive molecules, mainly in the form of food supplements for the active prevention of the main chronic degenerative diseases associated with aging. In addition, the interest is determined not only by the use of such plant-derived nutraceuticals in complementary medicine, in association or not with pharmacological products for the treatment of degenerative diseases, but also by the extraction methodology from the waste of agricultural production and industrial processing. The waste is reused through the recovery of active compounds, with a view toward a “circular” supply chain and zero waste economy for the development of potential innovative products. From this perspective, we have previously demonstrated the nutraceutical properties of two ancient Italian durum wheat husks belonging to the ancient cultivar “Senatore Cappelli” from two different cultivation areas (Puglia and Tuscany), assessing their potential as bioactive compound sources in terms of phytochemical, antioxidant, and anti-inflammatory properties [1]. In the present study, we have chosen to analyze the extracts obtained from Senatore Cappelli wheat, from seeds, chaff, flour, and pasta produced by it, as they constitute a sustainable supply chain. We aim at identifying bioactive molecules from the waste of some wheat product processing and develop extracts-enriched in molecules with a potential homeostatic role, counteracting inflammation and oxidative stress. We analyzed their anti-inflammatory as well as antioxidant activity in in vitro models by using BV2 murine microglia cell line and in in vivo models by evaluating antioxidant and immune-stimulating activity in *Caenorhabditis elegans* nematode. *C. elegans* has emerged as a powerful in vivo model organism for studying the biological effects of bioactive compounds due to its simplicity, well-characterized genetics, and physiological relevance. As a free-living nematode, *C. elegans* offers several key advantages for research, including its small size (~1 mm in adult length), short life span (~21 days for wild-type strains), and high reproductive output (~300 progeny via self-fertilization). These features make it an ideal system for high-throughput studies and for the rapid assessment of molecular pathways modulated by bioactive compounds. One of the most significant advantages of using *C. elegans* is its fully sequenced and well-annotated genome, which shares over 65% of its disease-related genes with humans [2]. Unlike traditional animal models, the use of *C. elegans* does not require the ethical approval of Institutional Animal Care and Use Committees, further streamlining research processes. Additionally, *C. elegans* can absorb compounds through multiple routes—ingestion, diffusion through the cuticle, and uptake via exposed sensory cilia—allowing for flexible experimental designs. The nematode’s digestive system facilitates rapid uptake and distribution of compounds, while its permeability barrier allows for diffusion into target tissues [3]. These characteristics make *C. elegans* an attractive alternative model for evaluating the bioactivity and mechanistic actions of various dietary and pharmacological agents. This investigation is critical as it provides insight into the potential health benefits of wheat extracts, particularly their roles in modulating oxidative stress and enhancing innate immune responses. Microglial cells are the main players in the development of neuroinflammation that underlies several chronic neurodegenerative diseases, including Parkinson’s disease and Alzheimer’s disease, and depression [4]. In these conditions, microglia polarizes towards a pro-inflammatory phenotype, M1, or classical activated phenotype that releases inflammatory cytokines and chemokines, showing a decreased phagocytic capability and an increased chemotactic activity. The process is induced and amplified by oxidative stress, targeting the transcription factors that bind to the promoter regions of inflammatory genes. For all these reasons, the modulation of microglia activity has recently been proposed as a therapeutic strategy to slow down the progression of neurodegenerative diseases. In the present study, we evaluated the functional phenotype of microglia stimulated by lipopolysaccharide (LPS) in the presence or in the absence of wheat extracts.

Indeed, to the best of our knowledge, this is the first study where the effects of extracts from wheat products have been assessed in terms of the potential antioxidant and innate immunity-stimulating effects using in vivo and in vitro model systems. Understanding these effects can prepare the way for the development of functional foods and therapeutic agents aimed at improving health outcomes.

## 2. Materials and Methods

### 2.1. Sampling

The Senatore Cappelli (SC) durum wheat samples were collected from crops cultivated on the karstic Murge upland (40°53′00″ N; 16°46′00″ E), a karstic plan located between Puglia and Basilicata (Figure 1), during the year 2020 as the growing season. The soil in this area is characterized by a high percentage of loam and clay, with a lower sand content, resulting in a flat terrain with good natural drainage, ideal for durum wheat cultivation. To prepare the soil, plowing was carried out in October, just a few days before sowing, ensuring optimal seedbed conditions. Sampling was conducted systematically at each step of the food production chain, encompassing seeds, flour, pasta, and chaff. To ensure representative-ness, approximately 3% of the total weight of each product was collected. Before analysis, both seeds and pasta were ground in liquid nitrogen to maintain the stability of biochemical compounds and prevent enzymatic degradation.

### 2.2. Hydroalcoholic Extraction of Durum Wheat Samples

Sample extraction followed the method of [5] with minor modifications. About 2 g (±0.01 g) of each “Senatore Cappelli” (SC) durum wheat sample (i.e., seeds, flour, and pasta) was weighed and placed in 15 mL glass centrifuge tube, and a volume of 5 mL ethanol aqueous solution (80:20, *v*/*v*) was added. The samples were homogenized using an ultrasonic and thermostatic bath (Bandelin Sonorex, RK100H, Berlin, Germany) at 400 MHz and room temperature for 5 min, followed by centrifugation at 3000 rpm for 10 min at room temperature (T = 25 °C) using a NEYA 10R refrigerated centrifuge (Exacta Optech, Modena, Italy). The supernatant was collected into a 10 mL flask. Another 5 mL of ethanol aqueous solution (80:20, *v*/*v*) was added to the residue, mixed, and centrifuged again for 10 min. The second extract was combined with the first and filled to a final volume of 10 mL with ethanol/water (80:20, *v*/*v*). The extracts were finally filtered through a 0.45 µm filter (Whatman^®^ Puradisc filters, Sigma Aldrich, Milan, Italy) before the analysis.

### 2.3. Total Phenolic Content (TPC)

The total phenolic content of wheat husk samples was determined using the Folin–Ciocâlteu method, as described by Vinci et al. [1]. Briefly, 1 mL of hydroalcoholic extract was mixed with 0.25 mL of Folin–Ciocâlteu reagent and 0.5 mL in a 10 mL amber volumetric flask. After 3 min, 0.5 mL of aqueous sodium carbonate solution (7.5% *w*/*v*) was added and the flask was kept in the dark for 30 min. Sample absorbance was measured at 750 nm against a blank solution (EtOH:H_2_O, 80:20 *v*/*v*). Results were expressed as milligrams of gallic acid equivalents per 100 g of wheat (mg GAE/100 g wheat samples) and obtained using a calibration curve ranging from 20 to 250 mg/L (R^2^ = 0.9998). All measurements were performed in triplicate.

### 2.4. Total Flavonoid Content (TFC)

The total flavonoid content of wheat samples was assessed using the aluminum chloride method, based on Abdel-Naeem et al. (2021), with slight modifications [6]. In a 5 mL volumetric flask, 0.5 mL of hydroalcoholic extract was mixed with 2 mL of distilled water and 150 μL of 5% NaNO_2_ (*w*/*v*), stirred, and incubated in the dark for 5 min. Then, 150 μL of 10% AlCl_3_ (*w*/*v*) was added, and the solution was incubated for another 5 min. Subsequently, 2 mL of 1 M NaOH was added, and the solution was left in the dark for 15 more minutes. The final volume was adjusted to 5 mL with distilled water. The absorbance was measured at 510 nm against a blank (EtOH_2_O, 80:20 *v*/*v*). Results were expressed as milligrams of rutin equivalents per 100 g of wheat (mg RE/100 g) using a linear calibration curve ranging from 50 to 1000 mg/L (R^2^ = 0.9995). Data were reported as means ± standard deviation (SD) from three replicates.

### 2.5. Antioxidant Activity Determination by ABTS and DPPH Assays

The antioxidant activity of wheat samples was evaluated using two spectrophotometric methods: DPPH and ABTS assays. For the DPPH assays, 1.5 mL of DPPH solution (2.5 ng/mL) was added to 1 mL of hydroalcoholic extract and kept in the dark for 30 min at room temperature, based on a previously reported method [1]. The free radical scavenging activity was measured at 517 nm. All experiments were performed in triplicate, and the results were expressed as the mean EC_50_ ± standard deviation (SD), where EC_50_ represents the concentration of wheat extract (mg/mL) that scavenged 50% of the radicals. Gallic acid standard, diluted in methanol (1–100 mg/L), was used as a positive control. The ABTS assay, based on the method by Pierre et al. (2015) with minor modifications [7], was used to determine the Trolox-equivalent antioxidant capacity (TEAC). In this assay, 0.4 mL of hydroalcoholic extract was mixed with 3.6 mL of ABTS radical solution and incubated in the dark for 15 min. The decrease in ABTS radical was measured by recording the absorbance at 734 nm. The results were expressed as milligrams of Trolox equivalents per 100 g of wheat sample (mg TE/100 g) based on a calibration curve ranging from 0.5 to 200 mg/TE (R^2^ = 0.9963).

### 2.6. Biogenic Amines Determination in Durum Wheat Samples

The determination of biogenic amines (BAs) was performed following a previously published method, with some modifications [1]. In brief, 0.40 g (±0.01 g) of SC wheat samples (seeds, flour, and pasta, respectively) was mixed with 5 mL of 0.6 M HClO_4_. The samples were homogenized and centrifuged at 3000 rpm for 10 min at 25 °C. The supernatant was collected, and the extraction process was repeated twice. The second extract was combined with the first and filtered through a 0.45 μm membrane filter. The final volume was adjusted to 10 mL with 0.6 M HClO_4_. For derivatization, a 1 mL aliquot of the final extract was mixed with 200 μL of 2 M NaOH, 300 μL of saturated NaHCO_3_ solution, and 2 mL of dansyl chloride solution (10 mg/mL in acetone). After stirring, the samples were left in the dark at 45 °C for 60 min. To stop the dansyl chloride reaction, approximately 100 μL of 25% NH_4_OH was added, and the final volume was brought to 5 mL with acetonitrile. The dansylated extract was filtered through a 0.45 μm membrane filter and injected into the HPLC system. The analytes were separated using a Supelcosil RP LC-18 column (250 mm × 4.6 mm; 5 μm) with a Supelguard LC-18 pre-column and fluorometric detection. The flow rate was set to 1.2 mL/min, and the column temperature was maintained at 30 °C. The elution began with 3 min of isocratic elution (50% acetonitrile, and 50% water), reaching 100% acetonitrile after 18 min. The initial isocratic condition was then restored. The method’s accuracy (recovery > 95%) and precision (RSD < 4.6%) were assessed by analyzing SC extracts at three different concentrations of BAs. The results from the triplicate analyses were expressed using a calibration curve for each BA, with a range from 0.1 to 25 mg/L (Table 1). In addition, the Biogenic Amines Quality Index (BAQI) was calculated using BAs results to assess the quality degradation of the SC samples. A BAQI value below 10 indicates that the product can be considered safe for consumption [8]. The BAQI was determined using the following formula and is expressed in mg/kg:$$B.A.Q.I.=\frac{\left(PUT+CAD+HIS\right)}{\left(1+SPM+SPD\right)}$$
where PUT represents putrescine, CAD represents cadaverine, HIS represents histamine, SPM represents spermine, and SPD represents spermidine.

### 2.7. NMR Spectroscopy

To determine the composition of the ethanolic extracts used for the various assays, three 2.5 mL aliquots of chaff, seed, flour, and pasta extracts were dried under a flow of nitrogen gas and then resuspended in 700 µL of D_2_O containing 3-(trimethylsilyl)-2,2,3,3-d 4-propionate sodium salt (TSP) at 2 mM concentration as a quantitative reference standard for subsequent NMR analysis. Resonance assignment was carried out on the basis of 2D experiments ^1^H-^1^H TOCSY and ^1^H-^13^C HSQC as well as literature data [9]. The ^1^H monodimensional spectra were acquired on a JNM-ECZ 600R (JEOL, Tokyo, Japan) spectrometer, equipped with a 14.09 Tesla magnet with an operating frequency for hydrogen of 600 MHz and a multinuclear head. The one-dimensional spectra were acquired with a given number of points equal to 64 k, a spectral amplitude of 15 ppm (corresponding to 9000 Hz), 128 scans and a recycling time of 7.7 s for a total of 15 s per scan to guarantee the complete relaxation of resonances. Bidimensional ^1^H-^1^H TOCSY experiments were acquired employing a data matrix of 8 k × 256 points, a spectral width of 15 ppm in both dimensions, 64 scans, a recycling time of 2 s and a mixing time of 80 ms. Bidimensional ^1^H-^13^C HSQC experiments were acquired using a data matrix of 8 k × 256 points, a spectral amplitude of 15 ppm for hydrogen and 250 ppm for carbon (corresponding to 37,500 Hz), 128 scans, a recycling time equal to at 2 s, and a direct J^1^H–^13^C coupling constant of 145 Hz.

### 2.8. In Vitro Tests

#### 2.8.1. Cytotoxicity Analysis by Trypan Blue Assay

The murine microglia cell line (BV2), graciously provided by Dr. Mangino of Sapienza University of Rome, was cultured in Dulbecco’s Modified Eagle’s Medium High Glucose (DMEM, D5671-500ML, Sigma Aldrich, St Louis, MO, USA) supplemented with 5% fetal bovine serum (FBS, F7524, Sigma Aldrich, St Louis, MO, USA), 1% L-glutamine, 1% penicillin-streptomycin, 1% nonessential amino acids, and 1% sodium pyruvate (Sigma Aldrich, St Louis, MO, USA). This was incubated at 37 °C in a humidified atmosphere with 5% CO_2_. BV2 cells were seeded in 48-well plates at a density of 30,000 cells per 400 µL. Extracts derived from grains, flour, and paste were administered at different concentrations (10, 100, 250, 500, 750, 1000 ng, 10, 50, 100 µg/mL), and the cells were incubated for 24 h at 37 °C. At the end of incubation, cells were detached using Tripsin-EDTA 1× (AU-L0940-100, Aurogene, Rome, Italy) and then counted using a Burker chamber in Trypan Blue (1:1) solution (25-900-CI, Corning, Glendale, AZ, USA). Both live and dead cells were counted.

#### 2.8.2. Real-Time qPCR

BV2 cells were seeded in 6-well plates at a density of 10^6^ cells per well in 1 mL of DMEM. After a 45 min pretreatment with husk extracts at concentrations of 10 and 100 ng/mL, the pro-inflammatory stimulus, LPS, was added at 1 ng/mL, and the cells were incubated at 37 °C for 4 h. At the end of this incubation, cells were lysed with 700 µL of Qiazol Lysis Reagent and stored at −80 °C. Total RNA was extracted from both control and treated cells using the miRNeasy Micro kit (Qiagen, Hilden, Germany) and quantified by NanoDrop One/OneC (Thermo Fisher Scientific, Waltham, MA, USA). The cDNA was synthesized using the high-throughput reverse transcription kit. Quantitative real-time PCR (qPCR) was performed for each sample in triplicate on an Applied Biosystems 7900HT fast real-time PCR system (Applied Biosystem, Cheshire, UK), using Power SYBR^®^ Green PCR Master Mix. Primers for real-time PCR amplification were designed through UCSC GENOME BROWSER https://genome.cse.ucsc.edu/ (accessed on 1 November 2022); University of California, Santa Cruz, CA, USA) (Table 2). Analysis of real-time PCR data was performed using the comparative threshold cycle (CT) method. The target quantity, normalized against the endogenous β-actin reference primer (ΔCT) and against the untreated control calibrator (ΔΔCT), was calculated by the 2^−ΔΔCT^ equation.

#### 2.8.3. Immunofluorescence

A quantity of 30,000 BV2 cells per well were plated in chamber slides in 200 μL of 10% DMEM FBS, and the cells were stimulated with wheat-derived extracts 100 ng/mL, LPS 1 ng/mL and IL4/10/13 20 ng/mL and incubated for 24 h. Three washes in PBS were performed and cells were fixed in 4% paraformaldehyde for 30′ RT. After 3 washes in PBS, BV2s were covered with TritonX-100 0.1% for 5′ and with Glycine 0.1 M for 20′ RT. After washes, cells were incubated with primary antibody 1:100 in PBS 4%BSA o.n. at 4 °C (anti-iNOS NB300-605 Novus Biologicals, anti-Arg1 AB-84248 (Immunological Sciences, Rome, Italy).

After washes, the secondary antibody (Rb CF488-a goat anti-Rb Ig(H+L) 20012, Biotium, Fremont, CA, USA) was added for 30′ RT (1:100 in PBS) in the dark. DAPI was added before closing the slide with VECTASHIELD Antifade Mounting Media (Vector Laboratories, Burlingame, CA, USA).

### 2.9. In Vivo Tests

#### 2.9.1. *C. elegans* Strains and Growth Conditions

The *C. elegans* strains used were wild-type N2, GST4::GFP transgenic strain, and skn-1 mutant. Nematodes were grown on nematode growth medium (NGM) and fed with heat-killed *Escherichia coli* OP50 and supplemented with 1:100 diluted wheat extracts. Afterward, 60 μL of heat-killed culture was spread on 3.5 cm diameter NGM plates and 100 μL of 1:100 diluted wheat extracts were added separately. Heat-killed OP50 cells were prepared as follows: bacteria were cultured overnight in Luria–Bertani (LB) broth at 37 °C, centrifuged at 6000 rpm for 15 min, and suspended in 2 mL of sterile water. Cells were then incubated at 65 °C for 90 min and deposited onto NGM agar plates. Heat-killed cells were also plated on LB agar in parallel to ensure that no viable cells remained.

#### 2.9.2. *C. elegans* Lifespan Assay

Synchronous nematodes were prepared as described in Schifano et al., 2021 [10] on NGM spread with heat-killed *E. coli* OP50 and 1:100 diluted wheat extracts. Nematodes grown on heat-killed *E. coli* OP50 were taken as controls. Lifespan analysis was performed at 16 °C and worms were transferred daily to new plates seeded with fresh lawns. They were scored as dead when they no longer responded to gentle touch with a platinum wire. At least 80 nematodes per condition were used in each experiment. All lifespan assays were performed in triplicate.

#### 2.9.3. Fluorescence Analysis in *C. elegans* Transgenic Strains

At the stage of 1 day of adulthood, synchronized gst-4::GFP transgenic worms fed heat-killed *E. coli* OP50 and supplemented with wheat extracts from embryo hatching were anesthetized with sodium azide (20 mmol L^−1^) (Sigma-Aldrich, St. Louis, MO, USA) and observed by Zeiss Axiovert 25 microscope (Zeiss, Oberkochen, Germany) as described in Bianchi et al., 2020 [11]. The experiments were repeated three times and 15 worms per group were used in each experiment. Images were taken at the time of exposure of 0.2 s and fluorescence was analyzed using ImageJ software 1.54k. Scale bars were inserted by Zeiss ZEN Microscopy Software 2011.

#### 2.9.4. Evaluation of Reactive Oxygen Species (ROS) Levels in *C. elegans*

ROS formation in 1-day adult worms, treated or not with wheat extracts from embryo hatching, was measured using the fluorescent probe H_2_DCFDA, as described in Yoon et al., 2018 [12] with some modifications. Briefly, worms were collected (in triplicate) in a 96-well microplate and washed in the M9 buffer. H_2_DCFDA (Sigma-Aldrich, Milan, Italy) probe was added in each sample to obtain a final concentration of 50 μM. To simulate oxidative stress, H_2_O_2_ was added, maintaining the concentration of 50 μM. After 1 h of dark incubation at 20 °C, worms were analyzed by using a microplate reader at excitation/emission wavelengths of 485 and 520 nm.

#### 2.9.5. RT-qPCR

RNA of 200 1-day adults supplemented or not with wheat extracts (at the concentration 2 mg/mL) from embryo hatching was extracted using miRNeasy Micro Kit (Qiagen), and real-time analysis with I Cycler IQ Multicolor Real-Time Detection System (Biorad, Hercules, CA, USA) was performed according to Schifano et al., 2019 [13]. The selective primers (200 nM) for genes daf-16, sek-1, pmk-1, gst-4, and sod-3 were reported in Table 2. Quantification was performed using a comparative *C*_T_ method (*C*_T_ = threshold cycle value). Briefly, the differences between the mean *C*_T_ value of each sample and the *C*_T_ value of the housekeeping gene (*act-1*) were calculated: Δ*C*_Tsample_ = *C*_Tsample_ − *C_ACT1_*. The result was determined as 2^−ΔΔCT^, where ΔΔ*C*_T_ = Δ*C*_Tsample_ − Δ*C*_Tcontrol_. The experiment was performed in triplicate.

## 3. Results

### 3.1. Phenolic and Antioxidant Properties of “Senatore Cappelli” Durum Wheat Samples

The total polyphenols (TPC), flavonoid contents (TFC), and antioxidant activity of “Senatore Cappelli” durum wheat samples (seeds, flour, pasta, and chaff) were evaluated through spectrophotometric assays (Table 3). Results highlight a variable content among the SC durum wheat chain analyzed samples. In particular, total polyphenols were found to be highest (*p* < 0.05) in the wheat chaff (164.17 ± 3.01 mg GAE/100 g), followed by seeds, pasta, and then flour showing the lowest total phenolic content. This observation aligns with previous studies showing that wheat by-products (such as husks and bran) exhibit higher TPC than other wheat fractions such as grains, flour, and pasta. However, processing into flour and pasta led to a significant decrease in TPC, thus highlighting a reduction of approximately −7.5% compared to raw matrices. Phenolics in flour mostly originate from the endosperm, which contains mostly soluble, low-molecular-weight compounds rather than the complex-bound phenolics concentrated in the bran and husk. The present findings, with flour showing 146.81 mg GAE/100 g, are consistent with literature reports highlighting up to a 60% reduction in TPC during the production of refined flour [14]. In detail, the production of refined flour involves separating the starchy endosperm from these outer layers, significantly lowering the phenolic content due to mechanical disruption and oxidation [14]. However, the TPC in pasta remains significantly lower than in chaff, underscoring the loss of phenolic-rich bran fractions during the initial milling process. In this regard, during pasta production, additional steps such as mixing, kneading, and high-temperature drying further degrade phenolic compounds due to oxidation and thermal breakdown. These processes not only remove phenolic-rich fractions but also alter the structure and extractability of remaining antioxidants, leading to an overall reduction in measured phenolic content [14,15,16]. In detail, the mechanisms that improve bioavailability during milling can be mainly attributed to the following: (i) cell wall disruption; (ii) particle size reduction—as particle size decreases, the exposure of phenolic compounds to enzymatic activity or extraction solvents increases, further improving their availability; (iii) enzymatic activation, e.g., polyphenol oxidases can become activated; (iv) thermal transformation (e.g., light heat can break ester or glycosidic bonds, releasing free phenolic acids, such as ferulic acid and other phenolics bound to dietary fibers), that may become more bioavailable under moderate thermal conditions; and (v) chemical changes, due to nonenzymatic reactions, such as the Maillard reaction, which can alter the chemical structure of phenolic compounds. Meanwhile, after milling, other factors contributing to improved bioavailability of phenolic compounds may be related to the following: i. formation of soluble phenolics, which can convert bound phenolics into their free or soluble forms; ii. increased extraction efficiency—the smaller particle size and larger surface area improve the efficiency of extraction processes, thereby improving the bioavailability of phenolic compounds; iii. synergistic effect together with other bioactive components, such as other macromolecules (proteins, starches) [15,16,17].

Similarly, the TFC followed a comparable trend, with the highest levels observed in chaff (136.21 ± 2.51 mg RE/100 g). Flavonoids tend to be concentrated in the outer layers of grains, and their reduction through processing into flour (125.14 ± 1.79 mg RE/100 g) and pasta (121.57 ± 2.13 mg RE/100 g) is expected at −6.0% and −9.0% compared to raw seeds.

Concerning antioxidant activity, the ABTS assay showed the highest antioxidant activity in chaff samples (12.30 ± 0.17 mg TE/100 g); moreover, it is worth noting the decrease of antioxidant fraction in the processed products, as flour and pasta compared to the raw seeds, thus highlighting the lower value of −11% and −18%, respectively. This particularly suggests the influence of milling and industrial processes on derived products. In particular, since a significant fraction of antioxidants is concentrated in the bran and germ layers, their removal during milling leads to a considerable reduction in the total antioxidant content. Consequently, refined flours, which consist mainly of the starchy endosperm, show lower antioxidant activity when analyzed by the ABTS assay. In addition, other factors such as mixing and kneading, high temperatures, and matrix effect can also affect the antioxidant activity of wheat products by (i) promoting oxidative reactions that degrade sensitive antioxidant compounds (e.g., phenolic acids and tocopherols); (ii) limiting the extractability of residual antioxidants during the ABTS assay, further lowering the apparent antioxidant activity; and/or (iii) causing thermal degradation of heat-sensitive antioxidants [15,16].

However, DPPH assays showed a different trend, with seed samples resulting in the highest antioxidant activity (EC_50_ of 7.74 ± 0.15 mg/100 g), while pasta showed the lowest antioxidant activity (EC_50_ of 2.29 ± 0.01 mg/100 g). This discrepancy between the ABTS and DPPH assays may be because these methods measure antioxidant activity differently from the DPPH radical, which bears no resemblance to peroxyl radicals involved in lipid peroxidation [18].

### 3.2. Biogenic Amines Content Among Senatore Cappelli Durum Wheat Samples

Table 4 shows the concentrations of the investigated biogenic amines in Senatore Cappelli wheat samples, along with the Biogenic Amines Quality Index (BAQI), which is used to assess product quality based on the biogenic amine profile.

The total BAs content slightly varied among the analyzed SC samples, with seeds highlighting the highest concentration (26.74 mg/kg) and chaff the lowest (6.55 mg/kg). Among investigated BAs, PUT and CAD were detected in all samples, with the highest concentration in seeds (2.39 ± 0.03 mg/kg; 0.15 ± 0.01 mg/kg, respectively). In detail, the PUT amount decreases in wheat-derived products by about −14% in flour and −16% in pasta compared to raw seeds. This suggests that processing may lower putrescine levels, possibly due to thermal degradation during pasta production [19,20]. HIS, TYR, and SER were not detected in any of the samples. In addition, naturally occurring polyamines such as SPD and SPM, known for their role in cell growth, were detected in all samples. Seeds had the highest levels of SPD (15.79 ± 0.25 mg/kg) and SPM (8.41 ± 0.16 mg/kg), reflecting their natural abundance in plant tissues [20]. These polyamines are naturally occurring compounds involved in cellular growth and development, and their content in seeds aligns with their role in germination and seed vitality. Their reduction in flour and pasta is worth noting due to the caryopsis breakdown or loss during milling and cooking, which reduces microbial activity and limits the formation of BAs [20,21]. In this sense, it is important to note that BAs can accumulate in foods through microbial activity or degradation of amino acids, especially during processing and storage [1]. Furthermore, the BAQI values for all samples analyzed: seeds, flour, pasta, and chaff (0.101; 0.097; 0.091, and 0.187, respectively) were below the safety threshold of <10 mg/kg, indicating that all samples are considered safe for consumption, therefore reflecting minimal quality loss due to biogenic amine accumulation.

### 3.3. Extracts Chemical Profile

In the examined spectra (chaff, seed, flour, and pasta ^1^H spectra are reported in Table 5), 31 molecules were identified and quantified, and their diagnostic resonances are reported in Table 5.

The identified molecules can be classified as amino acids, organic acids, carbohydrates and other molecules. It is interesting to highlight that several secondary metabolites, such as caffeic acid, gallic acid, and 4-hydroxybenzoic acid, as well as trimethylamine, were detected only in chaff. The results of the quantitative analysis are reported in Table 5.

### 3.4. Cytotoxicity of “Senatore Cappelli” Durum Wheat

Our previous results about cytotoxic activity of husk extracts on BV2 microglia cells assessed no toxicity at the concentrations of 100, 50, and 10 ng/mL, whereas at 100 and 10 ng/mL a significantly higher proliferation compared to untreated cells (CTRL) was detected [1].

Figure 2 shows the cytotoxicity induced by 24 h incubation of seeds, pasta, and flower extracts added at different amounts to BV2 microglia cells. Seed extracts do not show any toxicity until the concentration of 50 μg/mL, where a 22% decrease in the number of live cells compared to control cells (untreated cells) and a parallel increase in dead cells were detected. A further decrease in the number of live cells and a corresponding increase in the number of dead cells compared to the untreated cells were observed by increasing the concentration of seed extracts to 100 μg/mL.

By adding 100, 250, 500, and 750 ng/mL of flour extract, we noticed an increase of BV2 cell proliferation, with live cells doubling the number of the control cells after 24 h of incubation. A significant increase in dead cells was observed starting from the concentration of 50 μg/mL and a significant decrease in the number of live cells compared to the control starting from the concentration of 10 μg/mL (*p* < 0.038). This trend was also repeated with the pasta extracts, where, however, a cytotoxic effect appeared starting from the concentration of 50 μg/mL (Live cells: 54,500; Dead Cells: 16,000) and increasing (Live cells: 50,250; Dead Cells: 26,388) at the concentration of 100 μg/mL.

### 3.5. M1–M2 Polarization Markers Expression Following Incubation of BV2 Cells with Seeds, Flower, Pasta Extracts, and LPS Treatment

In order to evaluate BV2 cells pro-inflammatory/M1 or anti-inflammatory/M2 polarization after the addition of extracts obtained from wheat-derived products, we analyzed the IL-β, TNF-α, IL-6, IL-10, iNOS, and Arg-1 mRNAs expression after 45 min of pretreatment with the extracts, and then with or without the addition of 1 ng/mL LPS in the following 4 h of incubation, as shown in Figure 3 and Figure 4. We analyzed the activity of the extracts at two different concentrations, namely, 10 and 100 ng/mL, based on the results obtained with the cytotoxicity test. Flour extracts did not induce IL-1 β mRNA expression but, on the contrary, stopped its induction by LPS. Pasta extracts did not induce IL-1 β mRNA expression by itself, but they were not able to block the pro-inflammatory action of LPS even if they were shown to reduce it significantly (*p* < 0.012). Seed extracts behaved as flour extracts, blocking the expression of IL-1β mRNA induced by LPS addition. TNF-α mRNA was not or very weakly induced by flour, pasta, seed extracts, also in the presence of LPS, with the exception of the 100 ng/mL pasta extract, which failed to abolish the pro-inflammatory activity of LPS while significantly reducing it. All the examined extracts reduced the IL-6 mRNA expression induced by LPS but failed to completely decrease it. The mRNA of IL-10, an anti-inflammatory cytokine, was induced by the addition of flour 100 ng/mL, pasta 10 ng/mL, and seeds 100 ng/mL: all the extracts counteracted the LPS pro-inflammatory stimulation, inducing an increase of IL-10 mRNA.

As far as the expression of M1–M2 polarization markers is concerned, we analyzed the expression of iNOS and ARG-1 mRNAs, respectively. A very weak induction of iNOS was observed after treatment with flour extracts, whereas the addition of pasta and seed extracts did not induce the expression of this pro-inflammatory marker. On the contrary, all extracts inhibited the expression of iNOS mRNA following LPS addition, even with different degrees of significance. Flour and seed extracts promoted the expression of ARG-1 mRNA, a marker of anti-inflammatory polarization, flour both at 10 and 100 ng/mL, and pasta at 10 ng/mL also after LPS stimulation.

In order to confirm the activity of wheat-derived extracts on the expression of iNOS and Arg-1, we performed some immunofluorescence experiments in the presence or in the absence of 1 ng/mL LPS. As shown in Figure 5 and quantified by the plot, the extracts were all more effective than an anti-inflammatory cocktail in inducing Arg-1 and reducing iNOS expression, even in the presence of LPS.

Next we analyzed the expression of Nrf2, a transcription factor which, following the application of oxidative stress, dissociates from the Keap-1 factor, which keeps it in the cytoplasm and controls its degradation following ubiquitination. It then translocates into the nucleus, where it binds to ARE sequences, inducing the expression of around 200 genes, including SOD1 and GPX, enzymes with antioxidant activity. Therefore, the expression of NRF2 is associated with a protective activity. As depicted in Figure 6, seeds, flour, and pasta at both 100 and 10 ng/mL induce the expression of Nrf2. Similarly, they induce the expression of SOD1 and GPX in cells stimulated with LPS.

### 3.6. Antioxidant Response to Wheat Extract Supplementation in C. elegans

To assess the impact of wheat extracts on oxidative stress, reactive oxygen species (ROS) levels were measured in wild-type N2 *C. elegans* under both basal conditions and following exposure to hydrogen peroxide (H_2_O_2_). ROS are chemically reactive molecules containing oxygen that are by-products of cellular metabolism. While ROS are important for cellular signaling and homeostasis, their overproduction can lead to oxidative stress, damaging proteins, lipids, and DNA. To combat this oxidative stress, organisms have evolved various antioxidant defense systems. As shown in Figure 7, under basal conditions, no significant differences in ROS levels were observed between the control group and nematodes treated with the various wheat extracts, indicating that wheat extract supplementation alone does not substantially alter ROS levels in the absence of an external oxidative stressor. However, when nematodes were exposed to oxidative stress via H_2_O_2_ treatment, a notable reduction in ROS levels was observed in all wheat extract-treated groups compared to the H_2_O_2_-treated control. Specifically, ROS levels were reduced by approximately 30% in nematodes supplemented with each of the wheat extracts, suggesting that the extracts provided in vivo a protective effect against oxidative stress.

To further investigate the molecular basis of this protective effect, the expression of key oxidative stress-response genes, *gst-4* and *sod-3*, was analyzed via real-time qPCR. These genes are crucial for *C. elegans*’s defense against oxidative stress, with *gst-4* (glutathione S-transferase) playing a role in detoxifying ROS and *sod-3* (superoxide dismutase) neutralizing superoxide radicals. Altered expression levels of these genes indicate modifications in the organism’s antioxidant response mechanisms.

Real-time qPCR analysis revealed a significant reduction in *gst-4* expression in worms supplemented with various wheat extracts compared to the untreated control group, as shown in Figure 8. Specifically, *gst-4* expression decreased by 70% in samples treated with pasta, flour, and chaff extracts, while seed extract treatment led to a 50% reduction relative to the control. Similarly, *sod-3* expression showed a reduction of 50% or more across all wheat extract treatments compared to the untreated control.

Together, these findings suggest that although wheat extracts do not directly impact ROS levels under non-stressed conditions, they modulate the expression of oxidative stress-response genes and provide a notable protective effect against oxidative damage during H_2_O_2_-induced stress.

To evaluate the effects of wheat-derived extracts on the expression of GST-4, the transgenic *C. elegans* (*gst-4::GFP*) was employed using fluorescence microscopy on 1-day adults. Nematodes were treated with wheat seed, flour, pasta, and chaff extracts at a concentration of 2 mg/mL, and the mean fluorescence intensity (MFI) was measured to assess GST-4 expression levels. As shown in Figure 9A, the fluorescence microscopy images revealed differences in GST-4 expression across the various treatments. These differences were quantified by measuring MFI, and the results indicated a differential expression depending on the wheat extract used (Figure 9B).

Specifically, nematodes treated with wheat seeds and chaff showed no significant differences in GST-4 expression compared to the untreated control; in contrast, both flour and pasta extracts caused a slight but noticeable increase in GST-4 expression.

### 3.7. Innate Immunity Stimulation in C. elegans

*C. elegans* possesses a robust innate immune system that enables it to respond effectively to various stressors. The expression of key genes involved in the innate immune response and stress resistance pathways—*daf-16*, *sek-1*, and *pmk-1*—was evaluated by real-time qPCR in nematodes treated with wheat extracts. These genes are critical components of the IIS (insulin/IGF-1 signaling) and p38 MAPK pathways, which regulate immunity, stress responses, and longevity in *C. elegans* [22]. The IIS pathway begins with the activation of DAF-2, an insulin/IGF-like receptor [23]. This activation inhibits DAF-16, a key transcription factor responsible for promoting cellular defense under stress conditions by activating detoxification and stress-response genes. Conversely, when IIS is downregulated, as seen under stress conditions such as oxidative stress, DAF-16 translocates to the nucleus to induce the expression of genes involved in stress resistance and longevity.

Figure 10 shows that *daf-16* expression increased by 50% in nematodes treated with pasta extract, indicating a significant upregulation of this transcription factor, which may enhance stress resilience and immune defenses. Conversely, treatment with flour or chaff extracts resulted in *daf-16* expression comparable to the untreated control, suggesting a neutral effect on this pathway under these conditions. Interestingly, nematodes treated with seed extract showed an 80% reduction in *daf-16* expression, which could indicate a suppression of the IIS-mediated stress response, maybe activating other stress-response pathways.

Regarding *sek-1*, a key mediator in the p38 MAPK pathway [24] essential for regulating stress responses, we observed that wheat flour-treated nematodes exhibited expression levels similar to the control. In contrast, nematodes treated with pasta extract showed a 40% increase in *sek-1* expression, indicating an activation of the p38 MAPK pathway. The most notable effect was observed with chaff extract, which induced a two-fold increase in *sek-1* expression, suggesting a strong activation of this stress-response pathway. Nematodes treated with wheat seed extract showed a slight increase in *sek-1* expression compared to the control.

For *pmk-1*, a critical downstream effector of the p38 MAPK pathway involved in stress signaling [23], wheat flour-treated nematodes exhibited an 80% increase in expression, suggesting a robust activation of this immune pathway. Treatment with pasta, chaff, and seed extracts also led to elevated *pmk-1* levels, with increases of approximately 40%.

## 4. Discussion

There is a great interest in natural compounds for the development of new therapeutic strategies and, according to several studies, the combination of different bioactive compounds can lead to a better neuroprotection. It is known that a diet based on vegetables and fruit promotes health, and various substances derived from botanical extracts have shown antioxidant and anti-inflammatory properties, which help to prevent the development of diseases. Normally, however, the dosages of these compounds, if taken with food, do not reach sufficient concentrations to produce a very good effect. Therefore, these compounds extracted from foods and concentrated through various purification techniques have gone on to constitute those supplements then called nutraceuticals [25].

Numerous studies have confirmed that cereals exert a protective action on human health, and they are a rich source of bioactive components, including dietary fiber, proteins, and antioxidants, targeting cholesterol levels as well as inflammatory and pro-oxidant processes [26].

Dietary fiber and arabinoxylans of different durum (*Triticum turgidum* ssp. *Durum* L.) and bread (*Triticum aestivum* L.) wheat flours were analyzed and bioactive compounds (phenolics and tocopherols) were quantified, showing that whole flours contained higher total phenolics compounds and higher antioxidant properties [27]. Our results showed that total polyphenols were highest in the wheat chaff and raw seeds: this enrichment is primarily due to the localization of phenolic compounds in the outer layers of wheat, which are largely retained in the chaff during milling and processing [1,28]. For instance, Adom et al. (2005) report that wheat by-products exhibit up to three times the TPC of endosperm fractions, consistent with the higher values observed in wheat chaff compared to grains and processed products [29].

The evaluation of bioactive compounds content in “Senatore Cappelli” durum wheat samples provides valuable insights into the nutritional and safety aspects, thus underlying that food processing significantly affects the levels of polyphenols, flavonoids, and antioxidant capacity, with raw seeds maintaining the highest concentrations of these bioactive compounds. Flour showed a notable increase in antioxidant activity as measured by the ABTS assay, suggesting that certain processes may enhance bioavailability. However, thermal treatments, such as those involved in pasta production process, led to reduction in bioactive content and antioxidant potential. Furthermore, the milling and industrial processes of transforming flour into pasta introduces additional processing effects, such as heating and hydration, which can alter phenolic content. In this regard, it is worth noting that the reduction in polyphenol content during milling and pasta production could be attributed to the loss of bran and germ, where polyphenols are most concentrated [15,16]. Additionally, thermal processing such as the high temperatures used in pasta production, may contribute to the degradation of heat-sensitive polyphenolic compounds [16]. Despite these reductions, the remaining polyphenol content in both flour and pasta still reflects their contribution as sources of dietary antioxidants.

Concerning BAs content, while seeds exhibited the highest total BAs, all samples had BAQI values below the safety threshold, indicating that they remain safe for consumption throughout the food chain. Given the complexity of the extract composition, while it is possible to observe qualitative and quantitative differences among them, it is not possible to single out a specific molecule that could be responsible for the bioactivity of each extract. Therefore, in vivo and in vitro analysis are required to address this issue.

In a recent paper, it has been shown the anti-inflammatory and antioxidant properties of wheat husk derived from the Puglia and Tuscany regions [1]. However, it is well known that the phytochemical profile of plants is strongly influenced by their pedoclimatic growth conditions [17,19]). Therefore, it is necessary to evaluate how the behavior of the extracts could change according to the raw material origin.

Park et al. have shown the anti-inflammatory activity of the hull of γ-irradiated wheat mutant lines (*Triticum aestivum* L.) by adding the isolated samples to LPS-stimulated RAW 264.7 cells, evaluating their inhibitory effect on NO production and recommending the use of wheat-derived hull as a natural health-promoting product in food supplements [30].

Our experimental model takes advantage of a microglial cell line, in that microglia cells are the main players in the development of neuroinflammation underlying several chronic neurodegenerative diseases [31,32]. Quiescent microglia are characteristic of homeostatic conditions in the CNS; when disturbances occur, the microglia begins to produce inflammatory molecules undergoing a polarization state named M1. There is another polarization state called M2 in which repair mechanisms prevail with release of neurotrophic factors and removal of misfolded proteins. Indeed, M1 microglia induce inflammation and neurotoxicity, while M2 microglia induce anti-inflammatory activity and neuroprotection [33]. The suppression of excessive pro-inflammatory responses is expected to be a target for the treatment and prevention of neurodegenerative diseases. Thus, promoting microglia polarization shift from M1 to M2 phenotype may be a more prospective strategy in the therapy of neurodegenerative diseases.

Our results suggest that the extracts obtained from seeds, flour, and pasta derived from the ancient wheat Senatore Cappelli meet the requirements to be included in the bioactive molecules capable of polarizing microglia towards an M2 anti-inflammatory phenotype. In fact, the addition of the extracts significantly reduces the synthesis of pro-inflammatory molecules induced by the bacterial LPS and indeed promotes the synthesis of anti-inflammatory factors (IL-10) and the expression of M2 markers such as Arg-1, also reducing the expression of inflammatory markers such as iNOS. Furthermore, all the extracts induce the expression of Nrf2, a transcription factor for more than 200 genes, many of which encode antioxidant enzymes. The Nrf2–Keap1 axis exerts a protective effect in different disorders that show as main pathological mechanisms oxidative stress and inflammation, including cardiovascular disease and neurodegenerative disorders [25,34,35,36,37]. The activation of the Nrf2 pathway is very important for the control of inflammatory processes; as a matter of fact, one of the factors induced by Nrf2 is HO-1, which inhibits NFkB, a transcription factor upstream of the expression of pro-inflammatory factors. In this way, inflammatory processes are downregulated. This machinery has been described for Camptothecin, an extract of the plant *Camptotheca acuminate*, which reduces loss of neurons in the substantia nigra of the midbrain in LPS-injected mice and inhibits M1 polarization whereas promoting M2 polarization. This process was demonstrated to occur via the AKT/Nrf2/HO-1 and NF-kB signals in Parkinson’s disease mice experimental models [38]. In addition, bioactive peptides obtained from wheat bran were shown to exhibit great antioxidant activities against H_2_O_2_-induced oxidative stress in HepG2 cells by interfering with the Keap1–Nrf2 interaction [39]. Moreover, the antioxidant effects of theaflavin may be due to its capacity to increase dose-dependently the Nrf2 expression, reducing miRNA-128-3p levels in vivo and in vitro [40]. Similarly, it is possible that interfering bioactive components in the extracts may potentially act synergistically or through mechanisms independent of TPC and TFC, thus modulating the in vitro and in vivo activity, particularly their performance in response to LPS administration. In particular, literature findings suggest that several factors could modulate the immune-stimulating activity of wheat extracts. Khan et al. (2024) attribute specific phenolic classes, such as gallic acid and caffeic acid, which are concentrated in the outer layers of wheat, to contribute significantly to anti-inflammatory activity by modulating key inflammatory pathways (e.g., NF-κB inhibition) [41]. In addition, thermal processes and high-temperature drying during pasta production can degrade key phenolic compounds, thereby reducing their possible anti-inflammatory effects, particularly in promoting anti-inflammatory polarization and activating antioxidant pathways such as Nrf2 [17].

These results underscore the importance of processing methods and matrix composition in preserving the bioactivity of wheat-derived products. Further clarification on these points could be incorporated in future studies to better and comprehensively evaluate the correlation between the in vitro and in vivo antioxidant activity and immune response.

In *C. elegans*, the wheat extracts modulate the expression of key genes involved in the IIS and p38 MAPK pathways, with differential effects depending on the type of extract. The pronounced increase in *daf-16* expression with pasta extract and the strong upregulation of *sek-1* and *pmk-1* with chaff and flour extracts highlight the potential of wheat-derived products to stimulate innate immunity and enhance stress resilience in *C. elegans*. Regarding *daf-16*, the contrasting effects of pasta and seed extracts could be attributed to differences in their phytochemical profiles. As shown in Table 5, pasta extracts contain higher levels of malic acid, which may enhance the IIS-mediated stress response by promoting daf-16 activity [42]. In contrast, the high levels of amino acids such as tryptophane in seed extracts might suppress IIS signaling and redirect the stress response towards alternative pathways, since tryptophan-mediated beneficial effects in *C. elegans* was independent of DAF-16/FOXO and SKN-1/Nrf2 signaling [43]. For *sek-1*, the pronounced increase with chaff extract might be linked to its higher content of phenolic acids, such as caffeic and gallic acids, as these are known to activate stress-response pathways like p38 MAPK [44]. Similarly, the higher *pmk-1* expression observed with flour extracts could be due to its relatively balanced composition of antioxidants, which may synergistically enhance this pathway. However, although its content is similar to that of seeds, we cannot exclude the presence of trace compounds not detectable by NMR that may still exhibit immunostimulatory activity. These results suggest that wheat extracts, particularly from pasta, flour, and chaff, could activate key genetic pathways responsible for maintaining cellular homeostasis and protecting against oxidative stress. The present study has shown that wheat extracts present anti-inflammatory as well as antioxidant activities and may be considered as prospective positive health agents for the preparation of functional foods. The health benefits of whole wheat consumption can be partially attributed to wheat’s phytochemicals, including phenolic acids, flavonoids, alkylresorcinols, carotenoids, phytosterols, tocopherols, and tocotrienols [45]. Moreover, the valorization of by-products from agricultural and agro-industrial activities would also have significant implications in terms of circular economy.

## 5. Conclusions

Overall, the findings highlight the potential of durum wheat products as sources of dietary antioxidants and the importance of processing conditions in preserving their nutritional quality. Continued research into optimizing these processes can support the ongoing improvement of nutrient-rich, safe, and consumer-preferred wheat-based products that align with modern dietary trends. Low levels of toxic amines (e.g., CAD and PUT) and BAQI values across the chain support the safety and quality of “Senatore Cappelli” wheat products. The findings also suggest the potential value of chaff for targeted applications, provided that microbial activity is controlled to mitigate CAD levels.

## Figures and Tables

**Figure 1 nutrients-17-00302-f001:**
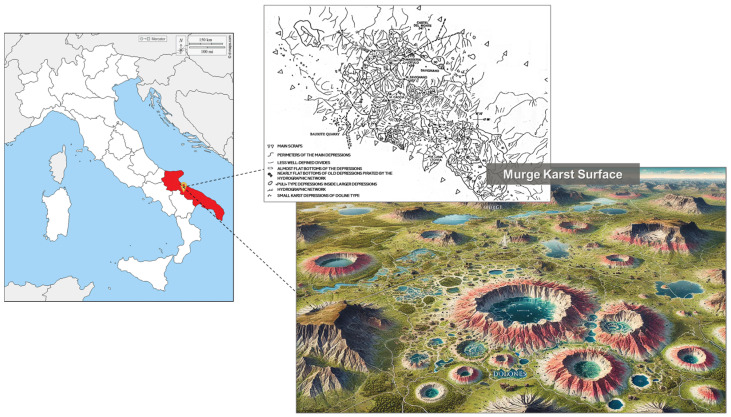
Geographical area of origin for the Senatore Cappelli durum wheat samples analyzed in the study.

**Figure 2 nutrients-17-00302-f002:**
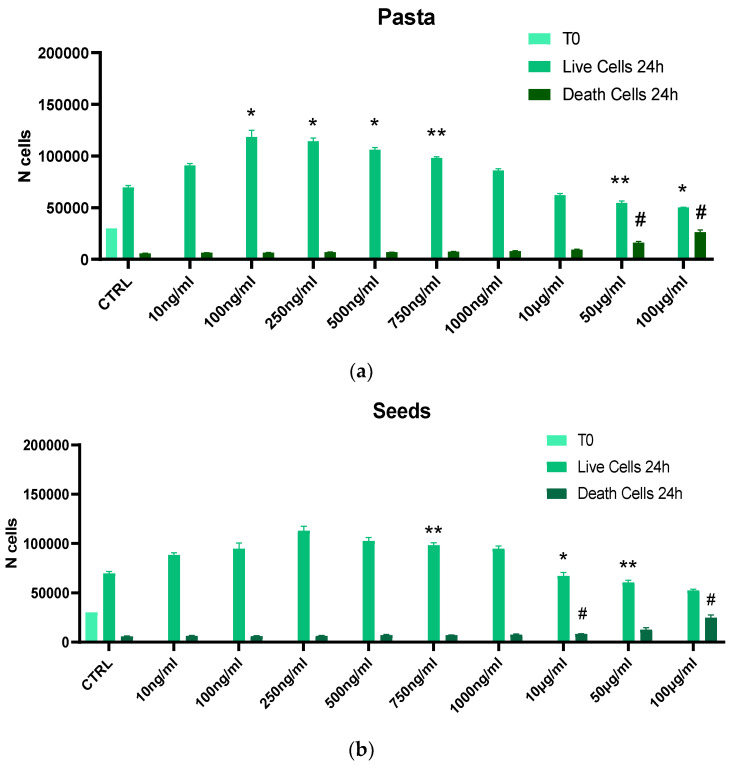
Trypan Blue assay for cytotoxicity of extracts on BV2 microglia cells. Cells were treated with 10, 100, 250, 500, 750, 1000 ng/mL and 10, 50, or 100 μg/mL extracts of pasta (**a**), seeds (**b**), and flour (**c**). Data are reported as mean ± SD and normalized to the control of at least three independent experiments, and statistical analysis was reported using unpaired Student’s *t*-test. * vs. Live Cells; # vs. Dead Cells; * *p* < 0.05; ** *p* < 0.01; # *p* < 0.05; ## *p* < 0.01.

**Figure 3 nutrients-17-00302-f003:**
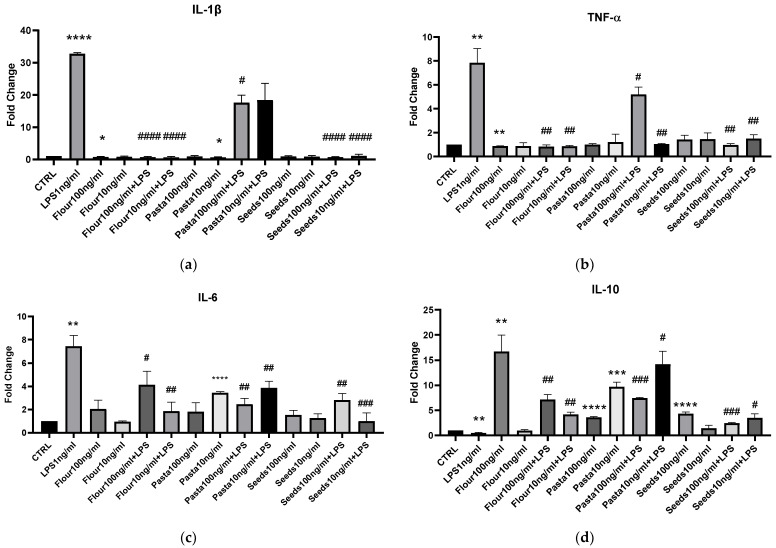
mRNA expression of IL-1β (**a**), TNF-α (**b**), IL-6 (**c**), and IL-10 (**d**) was evaluated by qRT-PCR. Data are shown as mean ± SD from three independent experiments performed in triplicate. Expression profiles were determined using the 2^−ΔΔCT^ method. Statistical analysis was reported using unpaired Student’s *t*-test. * vs. CTRL; # vs. LPS; * *p* < 0.05; ** *p* < 0.01; *** *p* < 0.001; **** *p* < 0.0001; # *p* < 0.05; ## *p* < 0.01; ### *p* < 0.001; #### *p* < 0.0001.

**Figure 4 nutrients-17-00302-f004:**
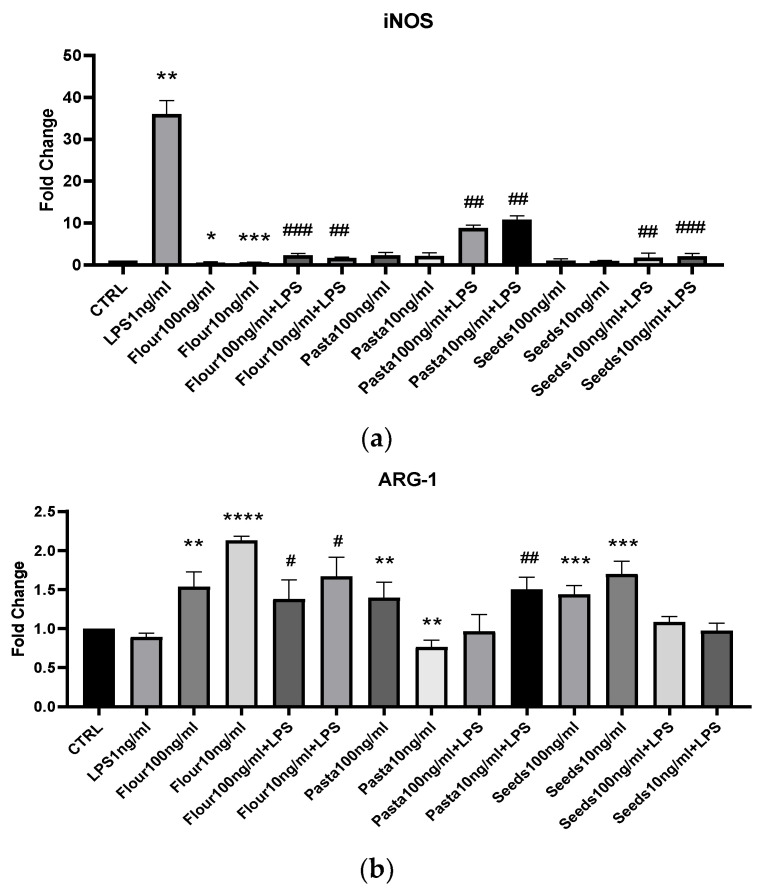
mRNA expression of iNOS (**a**) and ARG-1 (**b**) was evaluated by qRT-PCR. Data are shown as mean ± SD from three independent experiments performed in triplicate. Expression profiles were determined using the 2^−ΔΔCT^ method. Statistical analysis was reported using unpaired Student’s *t*-test. * *p* < 0.05; ** *p* < 0.01; *** *p* < 0.001; **** *p* < 0.0001; # *p* < 0.05; ## *p* < 0.01; ### *p* < 0.001.

**Figure 5 nutrients-17-00302-f005:**
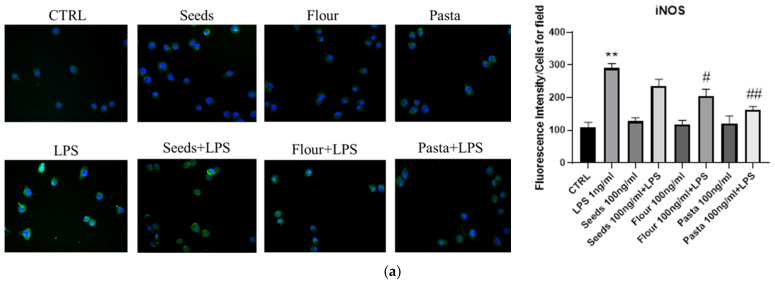
Immunofluorescence analysis of iNOS (**a**) and ARG-1 (**b**) expression in BV2 cells cultured with extracts in the presence or lack of LPS for 24 h. Quantification of the median fluorescence intensity was performed by ImageJ software and data were expressed as histograms, normalized to the number of cells for field. 4′,6-diamidino-2-phenylindole (DAPI) was used to counterstain the nuclei. Data are expressed as mean ± SD for each group (n = 3). Statistical analysis was performed by unpaired Student’s *t*-test. * vs. CTRL; # vs. LPS; * *p* < 0.05; ** *p* < 0.01; *** *p* < 0.001; # *p* < 0.05; ## *p* < 0.01; magnification: 20×.

**Figure 6 nutrients-17-00302-f006:**
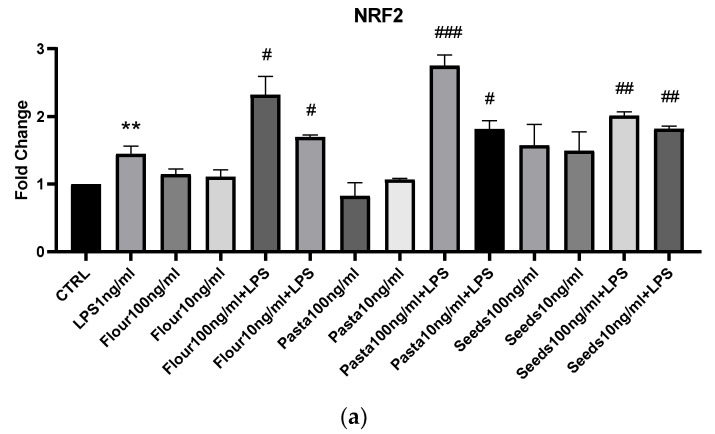
mRNA expression of Nrf2 (**a**), SOD1 (**b**), and GPX (**c**) was evaluated by qRT-PCR. Data are shown as mean ± SD from three independent experiments performed in triplicate. Expression profiles were determined using the 2^−ΔΔCT^ method. Statistical analysis was performed by unpaired Student’s *t*-test. * vs. CTRL; # vs. LPS; * *p* < 0.05; ** *p* < 0.01; *** *p* < 0.001; # *p* < 0.05; ## *p* < 0.01; ### *p* < 0.001.

**Figure 7 nutrients-17-00302-f007:**
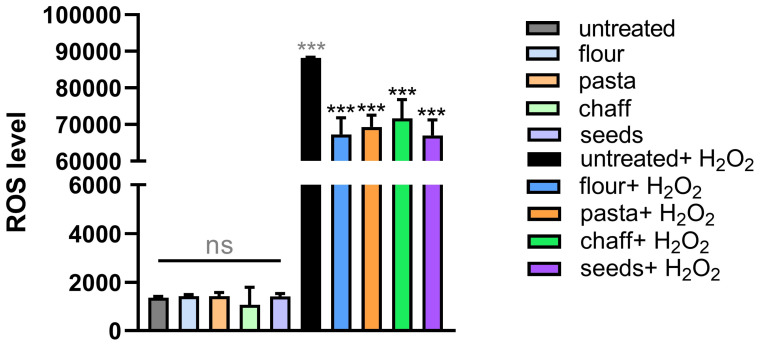
Effect of wheat extracts supplementation on *C. elegans* oxidative stress. Measurement of reactive oxygen species (ROS) levels in N2 worms supplemented with wheat extracts compared to untreated control and the same conditions in presence of H_2_O_2_. Experiments were performed in triplicate, and data are presented as mean ± SD (*** *p* < 0.001, ns: not significant).

**Figure 8 nutrients-17-00302-f008:**
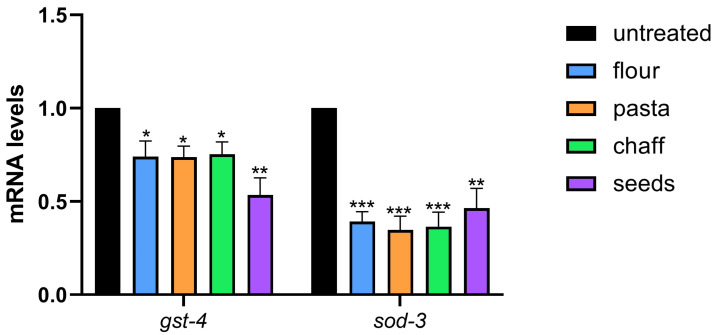
Expression of the detoxifying enzymes *sod-3* and *gst-4* genes in N2 worms treated with wheat extract (at the concentration 2 mg/mL) or untreated control on day 1 of adulthood. Histograms display gene expression related to oxidative stress, as detected by real-time PCR. Statistical analysis was evaluated by one-way ANOVA with the Bonferroni post-test; asterisks indicate significant differences (* *p* < 0.05, ** *p* < 0.01, *** *p* < 0.001). Bars represent the mean of three independent experiments.

**Figure 9 nutrients-17-00302-f009:**
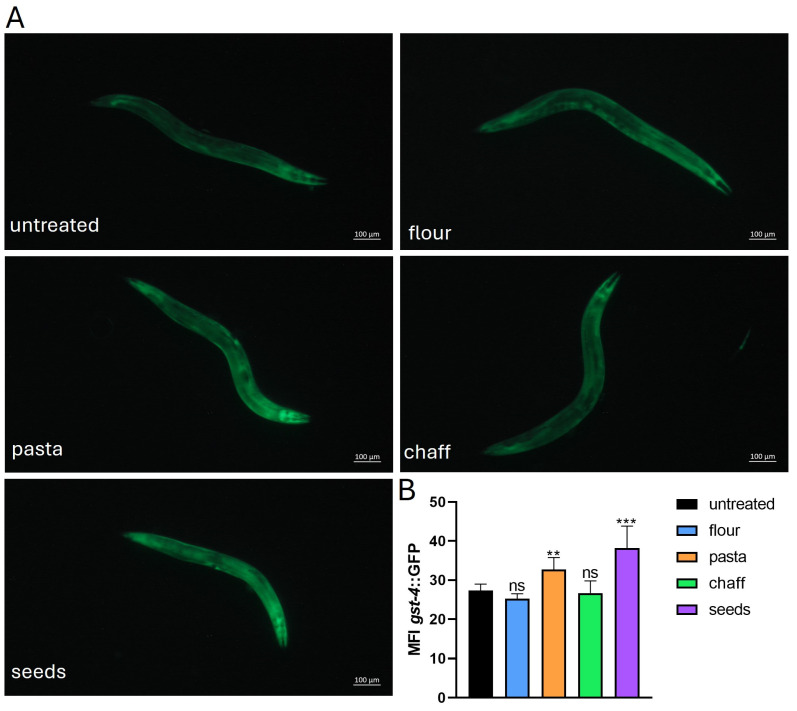
Analysis of oxidative stress in the transgenic *C. elegans gst-4*::GFP strain. (**A**) Fluorescence microscopy of 1-day adult nematodes treated with wheat extracts (at the concentration 2 mg/mL). Ten worms were considered for each condition. Scale bar = 100 μm. (**B**) Representation of the mean fluorescence intensity (MFI) of nematodes treated with the extracts. Statistical analysis was evaluated by one-way ANOVA with the Bonferroni post-test; asterisks indicate significant differences (** *p* < 0.01, *** *p* < 0.001, ns: not significant). Bars represent the mean of three independent experiments.

**Figure 10 nutrients-17-00302-f010:**
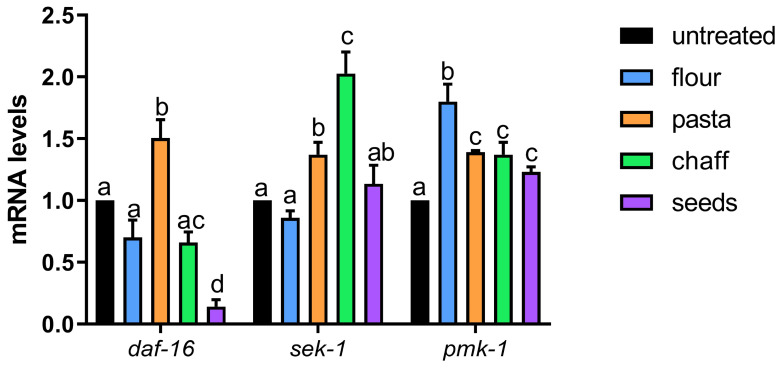
Real-time qPCR analysis of immune-related genes in *C. elegans*. Expression levels of daf-16, sek-1, and pmk-1 genes in 1-day adult nematodes treated with various 1:100 diluted wheat extracts. Experiments were performed in triplicate. Data are presented as mean ± SD. Different letters indicate *p* < 0.05.

**Table 1 nutrients-17-00302-t001:** Chromatographic parameters of biogenic amines analyzed by HPLC-UV/FD.

Biogenic Amines	Linear Range (mg/L)	Regression Eq.	Correlation Coefficient (R^2^)	RSD (%)	LOD (mg/L)	LOQ (mg/L)
B-PEA	0.1–25	y = 8 × 10^6^x − 51,893	0.999	1.32	0.04	0.12
PUT	0.1–25	y = 3 × 10^7^x − 177,123	0.997	1.02	0.03	0.09
CAD	0.1–25	y = 1 × 10^7^x − 116,338	0.997	1.06	0.02	0.07
HIS	0.1–25	y = 632,568x − 7223.8	0.998	2.71	0.08	0.24
SER	0.2–8	y = 73,089x +28,843	0.996	0.05	0.07	0.23
TYR	0.1–25	y = 1 × 10^6^x + 4302.6	0.996	0.71	0.30	0.96
SPD	0.1–25	y = 1 × 10^7^x − 85,162	0.997	1.14	0.05	0.15
SPM	0.1–25	y = 1 × 10^7^x + 24,850	0.997	1.89	0.09	0.29

Note: B-PEA, B-phenylethylamine; SER, serotonin; TYR, tyramine; PUT, putrescine; CAD, cadaverine; HIS, histamine; SPD, spermidine; SPM, spermine; LOD, limit of detection; LOQ, limit of quantification.

**Table 2 nutrients-17-00302-t002:** Primers used in real-time qPCR analysis.

Gene	Forward Primer (5′–3′)	Reverse Primer (5′–3′)	Accession Number
*mARG1*	ATGTGCCCTCTGTCTTTTAGGG	CTCTCACGTCATACTCTGT	NM_007482.3
*miNOS*	GGCAGCCTGTGAGACCTTTG	GCATTGGAAGTGAAGCGTTTC	AF427516.1
*mACT-β*	GGCTGTATTCCCCTCCATCG	CCAGTTGGTAACAATGCCATGT	NM_007393.5
*mSOD1*	GCCCGCTAAGTGCTGAGTC	AGCCCCAGAAGGATAACGGA	NM_017050
*mNRF2*	TCTGAGCCAGGACTACGACG	GAGGTGGTGGTGTCTCTGC	NM_031789
*mIL-1β*	GAAATGCCACCTTTTGACAGTG	TGGATGCTCTCATCAGGACAG	NM_008361.4
*mTNF-α*	CTGAACTTCGGGGTGATCGG	GGCTTGTCACTCGAATTTTGAGA	BC137720.1
*mIL-10*	GCCCTTTGCTATGGTGTCCTTTC	TCCCTGGTTTCTCTTCCCAAGAC	NM_010548.2
*mIL-6*	CGGAGAGGAGACTTCACAGAGGA	TTTCCACGATTTCCCAGAGAACA	NM_001314054.1
*mGPX*	AGGGTAGAGGCCGGATAAGG	CGAGCAGCACACATACTGGA	NM_008160
*sek-1*	CAGAGCCGTTTATTGGGAA	TGCATCCGGCTTGTACAGT	AB060731.2
*pmk-1*	AAATGACTCGCCGTGATTTC	CATCGTGATAAGCAGCCAGA	NM_068964.7
*daf-16*	TCAAGACCTCAAAGCCAAT	ACGAGAAAGAAGGAGTAAG	AF032112.1
*Act-1*	GAGCGTGGTTACTCTTTCAC	CAGAGCTTCTCCTTGATGTC	NM_073418.9

**Table 3 nutrients-17-00302-t003:** Total phenolic (TPC), flavonoid (TFC) content, and antioxidant activity by ABTS and DPPH assays in “Senatore Cappelli” durum wheat samples.

Samples	TPC (mg GAE/100 g)	TFC (mg RE/100 g)	ABTS Assay (mg TE/100 g)	DPPH Assay (EC_50_ in mg/100 g)
Seeds	159.58 ± 2.02 *	133.71 ± 1.44 *	11.28 ± 0.24 *	7.74 ± 0.15 *
Flour	146.81 ± 2.33 *	125.14 ± 1.79 *	10.05 ± 0.89 *	3.19 ± 0.09 *
Pasta	148.13 ± 1.21 *	121.57 ± 2.13 *	9.23 ± 0.12 *	2.29 ± 0.01 *
Chaff	164.17 ± 3.01 *	136.21 ± 2.51 *	12.30 ± 0.17 *	4.86 ± 0.02 *

TPC: Total Phenolic Content; TFC: Total Flavonoid Content; GAE: Gallic Acid Equivalent; RE: Rutin Equivalent; TE: Trolox Equivalent; EC_50_: extracts’ concentration providing 50% of radicals scavenging activity. The subscript * denotes significant differences (*p* < 0.05) in the same column.

**Table 4 nutrients-17-00302-t004:** Biogenic amines content in Senatore Cappelli durum wheat samples. Data are expressed as mg/kg of wheat sample ± standard deviation of n = 3 measurements.

Biogenic Amines Content (mg/kg)										
Sample	BPEA	PUT	CAD	HIS	SER	TYR	SPD	SPM	Tot BAs	BAQI
Seeds	<LOD	2.39 ± 0.03 *	0.15 ± 0.01 *	<LOD	<LOD	<LOD	15.79 ± 0.25 *	8.41 ± 0.16 *	26.74	0.101
Flour	<LOD	1.34 ± 0.01 *	0.14 ± 0.03 *	<LOD	<LOD	<LOD	9.49 ± 0.02 *	4.66 ± 0.02 *	15.36	0.097
Pasta	<LOD	1.06 ± 0.02 *	0.16 ± 0.01 *	<LOD	<LOD	<LOD	8.51 ± 0.07 *	3.97 ± 0.11 *	13.69	0.091
Chaff	<LOD	0.83 ± 0.03 *	0.36 ± 0.01*	<LOD	<LOD	<LOD	3.73 ± 0.14 *	1.63 ± 0.03 *	6.55	0.187

β-PEA: β-phenylethylamine; SER: serotonin; TYR: tyramine; PUT: putrescine; CAD: cadaverine; HIS: histamine; SPD: spermidine; SPM: spermine Total BAs: Total amount of biogenic amines; B.A.Q.I.: Biogenic Amines Quality Index. The subscript * denotes significant differences (*p* < 0.05) in the same column.

**Table 5 nutrients-17-00302-t005:** Quantitative analysis of the ethanol extracts measured by NMR spectroscopy. Statistical significance was assessed through ANOVA and Holm–Sidak tests.

	Molecule	Amount (mg/100 g of Dried Extract)			
		Chaff	Seed	Flour	Pasta
Amino Acids	Leucine	1.23 ± 0.11 a	0.20 ± 0.06 b	0.21 ± 0.01 b	0.645 ± 0.03 c
	Isoleucine	1.37 ± 0.07 a	0.18 ± 0.01 b	0.18 ± 0.01 b	0.64 ± 0.03 c
	Valine	2.19 ± 0.11 a	0.52 ± 0.03 b	0.34 ± 0.02 c	0.92 ± 0.05 d
	Threonine	9.49 ± 0.48 a	1.32 ± 0.07 b	0.58 ± 0.03 c	0.88 ± 0.05 d
	Alanine	8.63 ± 0.43 a	1.16 ± 0.06 b	0.76 ± 0.04 c	1.47 ± 0.08 d
	GABA	3.09 ± 0.16 a	0.12 ± 0.01 b	0.06 ± 0.01 c	2.49 ± 0.13 a
	Glutamate	3.34 ± 0.17 a	1.57 ± 0.08 b	1.07 ± 0.06 c	0.46 ± 0.03 c
	Glutamine	2.85 ± 0.14 a	0.20 ± 0.01 b	0.09 ± 0.01 c	0.29 ± 0.02 d
	Asparagine	2.11 ± 0.11 a	2.59 ± 0.13 a	1.07 ± 0.06 b	1.24 ± 0.06 b
	Tyrosine	1.99 ± 0.11 a	1.09 ± 0.06 b	0.50 ± 0.03 c	0.60 ± 0.03 c
	Phenylalanine	3.36 ± 0.17 a	0.77 ± 0.04 b	0.75 ± 0.04 b	1.76 ± 0.09 c
	Tryptophan	1.33 ± 0.07 a	6.02 ± 0.31 b	2.56 ± 0.13 c	3.34 ± 0.17 d
Organic acids	3-hydroxybutyric acid	2.17 ± 0.11 a	0.34 ± 0.02 b	0.57 ± 0.03 c	0.72 ± 0.04 d
	Acetic acid	7.11 ± 0.36 a	0.78 ± 0.04 b	0.45 ± 0.02 c	0.53 ± 0.03 c
	Succinic acid	2.27 ± 0.12 a	0.29 ± 0.02 b	0.47 ± 0.03 c	0.91 ± 0.05 d
	Malic acid	6.66 ± 0.34 a	1.73 ± 0.09 b	1.91 ± 0.11 b	5.97 ± 0.31 a
	Caffeic acid	0.17 ± 0.01 a	N.D.	N.D.	N.D.
	Fumaric acid	0.42 ± 0.02 a	0.24 ± 0.01 b	0.15 ± 0.02 c	0.78 ± 0.04 d
	Gallic acid	0.18 ± 0.01 a	N.D.	N.D.	N.D.
	4-hydroxybenzoic acid	0.17 ± 0.01 a	N.D.	N.D.	N.D.
	Formic acid	0.74 ± 0.04 a	0.24 ± 0.01 b	0.24 ± 0.01 b	0.32 ± 0.02 c
Carbohydrates	Sucrose	176.8 ± 8.8 a	271.7 ± 13.6 b	185.1 ± 9.3 a	198.4 ± 9.9 a
	Raffinose	42.58 ± 2.13 a	145.0 ± 7.3 b	61.2 ± 3.1 c	48.7 ± 2.5 a
	Glucose	34.2 ± 1.7 a	4.60 ± 0.23 b	8.59 ± 0.43 c	258.9 ± 12.9 d
	Threalose	11.16 ± 0.56 a	1.81 ± 0.09 b	0.43 ± 0.02 c	2.41 ± 0.12 d
Other molecules	Fatty Acids	43.26 ± 2.17 a	4.95 ± 0.25 b	4.54 ± 0.23 b	3.88 ± 0.21 b
	Trimethilamine	0.08 ± 0.01 a	N.D.	N.D.	N.D.
	Choline	8.78 ± 0.44 a	4.45 ± 0.22 b	2.14 ± 0.11 c	7.51 ± 0.38 a
	Betaine	24.92 ± 1.25 a	29.8 ± 1.49 a	17.26 ± 0.86 b	18.13 ± 0.91 b
	Uracile	0.42 ± 0.02 a	0.13 ± 0.01 b	0.09 ± 0.02 b	0.88 ± 0.11 c
	Trigonelline	0.90 ± 0.05 a	0.25 ± 0.01 b	0.21 ± 0.01 b	0.16 ± 0.01 c

N.D.: not detected. Different letters for every variable signify that the groups are statistically different (*p* < 0.05) among them according to ANOVA.

## Data Availability

The original contributions presented in this study are included in the article. Further inquiries can be directed to the corresponding author.

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
