# Peer review of "In Vitro and In Vivo Antioxidant and Immune Stimulation Activity of Wheat Product Extracts"

_nutrients, 2025, doi:10.3390/nu17020302_

Round 1

Reviewer 1 Report

Comments and Suggestions for Authors

The manuscript entitled “In vitro and in vivo antioxidant and immune stimulation activity of wheat product extracts” by Beatrice Mengoni et al examined the antioxidant immune response of various parts (seed, flour, chaff, pasta) of wheat and discussed the differences of their biological activities.

They used the hydroalcoholic extraction method (ethanol/water 80:20), from the chemical standpoint of view, there are many methods to extract biological substances from the plant materials, in this meaning, it is of interest to use the different the extraction method in the future.

The authors carried out many experiments and it is quite important to find the real active biological substrate and the part. However, the biological activities are different in many cases, it is quite difficult to focus on the one part. Some of these come from the many different types of experiments and put all these experimental results together.

They carried out many experiments, however, the discussion part in the manuscript is not so long. The authors quoted other references, which is quite important, however, they should discuss the relation of each experimental result. For example, the antioxidant activity and immune response, antioxidant activity and the amount of flavonoid content, there are many things to discuss based on their original experimental results.

Tables

In Table 3, The authors discussed the radical scavenging activity and TPC and TFC content. The figures used in the manuscript are different from the figures of Table 3. (lines 337-345) It is quite important point to clarify. In comparison of the total amount of TPC and TFC of seed and chaff, the values are larger chaff than seeds. However, the DPPH radical scavenging activity is not in this order. Some other factors might be responsible for this, it is desirable to respond this point.

In Table 5, the authors reported many biological samples detected by NMR spectroscopy and quantized the amount of each substrates. This information is quite important to discuss in the future project. For example, many compounds are only found in chaff. (caffeic acid, gallic acid, 4-hydroxybenzoic acid and trimethiamine). In the case of amino acids, chaff showed the highest amino acid amount in many cases, while flour showed the lower amount in almost all cases. Similar experimental results are also found in organic acids. If possible, the authors give some explanation for these points.

Figures

In Figure 3, the authors examined the inflammatory response of seeds, flour and pasta with or without administration of LPS, the trend of seeds and flour showed a similar behavior but pasta showed the low (no) inhibitory activity toward LPS administration. It is not obvious of the inflammatory response and TPC and TFC values (Table 3) and the compounds responsible for these responses. All these in vitro works are found in Fig. 4-6.

In Figure 7 and 8, the authors examined Ros level and antioxidant enzyme expression of C. elegans, all these results showed moderate antioxidant defense system was induced by the addition of the extracts. The difference of the extracts are not corresponding with  the of TPC and TFC values reported in Table 3.

In Figure 10, the authors showed the interesting results in the case of daf-16 mRNA expression, namely, the administration of pasta extract increase the mRNA level 50%, while the administration of seed extract reduced the mRNA expression 80%. This difference is important to discuss, although the authors briefly write as follow “,which could indicate a suppression of the IIS-mediated stress response, maybe activation other stress responses”(line 563-565). In the case of sek-1, chaff extract showed the highest increase, and in the case of pmk-1 flour showed the highest increase. These results demonstrate that the different compound is responsible for each response. It is quite meaningful to discuss these points with the compound list on Table 5.

The authors carried out many experiments and it is quite important for further studies, however, the experimental results well not well-organized enough. More precise explanation and discussion based on the experimental results are required in the revised version.

Comments on the Quality of English Language

There are some primitive spelling mistakes, which are easily revised by the use of word checking system. In general, it is not difficult to read the manuscript (quality is (very) good).

Author Response

The manuscript entitled “In vitro and in vivo antioxidant and immune stimulation activity of wheat product extracts” by Beatrice Mengoni et al examined the antioxidant immune response of various parts (seed, flour, chaff, pasta) of wheat and discussed the differences of their biological activities.

They used the hydroalcoholic extraction method (ethanol/water 80:20), from the chemical standpoint of view, there are many methods to extract biological substances from the plant materials, in this meaning, it is of interest to use the different the extraction method in the future.

The authors carried out many experiments and it is quite important to find the real active biological substrate and the part. However, the biological activities are different in many cases, it is quite difficult to focus on the one part. Some of these come from the many different types of experiments and put all these experimental results together.

They carried out many experiments, however, the discussion part in the manuscript is not so long. The authors quoted other references, which is quite important, however, they should discuss the relation of each experimental result. For example, the antioxidant activity and immune response, antioxidant activity and the amount of flavonoid content, there are many things to discuss based on their original experimental results.

Tables 

In Table 3, The authors discussed the radical scavenging activity and TPC and TFC content. The figures used in the manuscript are different from the figures of Table 3. (lines 337-345) It is quite important point to clarify. In comparison of the total amount of TPC and TFC of seed and chaff, the values are larger chaff than seeds. However, the DPPH radical scavenging activity is not in this order. Some other factors might be responsible for this, it is desirable to respond this point. 

The authors greatly thank the reviewer for his appreciable comments, and apologize for possible misunderstandings generated by the non-correspondence of the results entered in the table with those discussed in the manuscript. The authors have now completely revised this part, providing the correct information. In particular, in lines 334-340, discussions regarding the influence of processing on total polyphenol content have been expanded. In addition, the results regarding the antioxidant fraction have been completely revised and rewritten in lines 345-365, placing special emphasis on the factors that can affect the antioxidant activity of wheat products compared to raw materials.

In Table 5, the authors reported many biological samples detected by NMR spectroscopy and quantized the amount of each substrates. This information is quite important to discuss in the future project. For example, many compounds are only found in chaff. (caffeic acid, gallic acid, 4-hydroxybenzoic acid and trimethiamine). In the case of amino acids, chaff showed the highest amino acid amount in many cases, while flour showed the lower amount in almost all cases. Similar experimental results are also found in organic acids. If possible, the authors give some explanation for these points. 

Chaff, seed and flour are different entities from a botanical point of view. Indeed, chaff is the tegument of the seed, therefore it is enriched in phenols with a potential antimicrobial activity (i.e. gallic acid and 4-hydroxybenzoic acid) while there is a rather small water content. For seed, we consider the portioned by the germ and the endosperm, rich in terms of proteins (not recovered by the ethanolic extract) and other nutrients for the embryo. Flour is processed only from the endosperm and not the whole grain, therefore a difference in the content of both amino and organic acids is to be expected.

Moreover, pasta is derived from an industrial process in the presence of water that could increase some enzymatic reactions and consequently the number of free amino acids, the only ones to be detected by the method used

Figures

In Figure 3, the authors examined the inflammatory response of seeds, flour and pasta with or without administration of LPS, the trend of seeds and flour showed a similar behavior but pasta showed the low (no) inhibitory activity toward LPS administration. It is not obvious of the inflammatory response and TPC and TFC values (Table 3) and the compounds responsible for these responses. All these in vitro works are found in Fig. 4-6.

The authors would like to thank the reviewer for this comment which provided the opportunity to discuss and correlate the results of the work more comprehensively. Figure 3 shows the mRNA expression of pro- and anti-inflammatory cytokines. Pasta at 100ng/ml  reduces by 50% the IL-1beta mRNA expression, whereas at 10ng/ml the difference is not significant, probably for the high standard deviation Pasta  is effective in reducing TNF-alpha mRNA expression, in particular, a 40% reduction was observed with a concentration of 100ng/ml and a 90% reduction with a concentration of 10ng/ml. In inhibiting the expression of IL-6 mRNA, pasta behaves similarly to seeds and flour. This anti-inflammatory activity agrees the content of TPC and TFC, respectively. In addition all samples induce the expression of Arg-1 which is a marker of M2 polarization of innate immunity cells and down-regulate the expression of iNOS, a marker of M1 polarized innate immunity cells. These results are congruent with TPC (Total Complex Phenols) and TFC (Total Complex Flavonoids) values obtained by spectrophotometry for pasta, suggesting that the loss of phenol-rich bran fractions during the initial milling process could explain the difference with the raw matrices.

We believe that our data underline the possibility that bioactive compounds derived from Senatore Cappelli wheat and its processed products can counteract inflammatory reactions, especially chronic low-grade ones that insidiously increase the risk of developing chronic-degenerative diseases

Following the reviewer's notes, whom we thank, an entire paragraph focused on this issue has been added (lines 841-866)

In Figure 7 and 8, the authors examined Ros level and antioxidant enzyme expression of C. elegans, all these results showed moderate antioxidant defense system was induced by the addition of the extracts. The difference of the extracts are not corresponding with the of TPC and TFC values reported in Table 3.

We thank the Reviewer and we agree with the comment. While all the extracts demonstrated antioxidant activity in C. elegans under induced stress conditions, the lack of correspondence between the TPC and TFC values and the in vivo response may be due to the presence of other bioactive components in the extracts that can influence the antioxidant response. These components, potentially acting synergistically or through mechanisms independent of TPC and TFC, could modulate the in vivo effects.

In Figure 10, the authors showed the interesting results in the case of daf-16 mRNA expression, namely, the administration of pasta extract increase the mRNA level 50%, while the administration of seed extract reduced the mRNA expression 80%. This difference is important to discuss, although the authors briefly write as follow “,which could indicate a suppression of the IIS-mediated stress response, maybe activation other stress responses”(line 563-565). In the case of sek-1, chaff extract showed the highest increase, and in the case of pmk-1 flour showed the highest increase. These results demonstrate that the different compound is responsible for each response. It is quite meaningful to discuss these points with the compound list on Table 5. 

We thank the Reviewer for the comment and agree that the observed differences in daf-16, sek-1, and pmk-1 expression are significant and need a deeper discussion. The manuscript has been modified as follows " Regarding daf-16, the contrasting effects of pasta and seed extracts could be attributed to differences in their phytochemical profiles. As shown in Table 5, pasta extracts contain higher levels of malic acid, which may enhance the IIS-mediated stress response by promoting daf-16 activity 10.1371/journal.pone.0058345. In contrast, the high levels of amino acids such as tryptophane in seed extracts might suppress IIS signaling and redirect the stress response towards alternative pathways, since tryptophan-mediated beneficial effects in C.elegans  was independent of DAF-16/FOXO and SKN-1/Nrf2 signaling 10.1186/s12863-015-0167-2. For sek-1, the pronounced increase with chaff extract might be linked to its higher content of phenolic acids, such as caffeic and gallic acids, as these are known to activate stress response pathways like p38 MAPK  10.1007/s10522-011-9334-7, 10.1016/j.fochx.2024.101233. Similarly, the higher pmk-1 expression observed with flour extract could be due to its relatively balanced composition of antioxidants, which may synergistically enhance this pathway. However, although its content is similar to that of seeds, we cannot exclude the presence of trace compounds not detectable by NMR that may still exhibit immunostimulatory activity.”

The authors carried out many experiments and it is quite important for further studies, however, the experimental results well not well-organized enough. More precise explanation and discussion based on the experimental results are required in the revised version. There are some primitive spelling mistakes, which are easily revised by the use of word checking system. In general, it is not difficult to read the manuscript (quality is (very) good).

We really thank the reviewer for his comments We included two new paragraphs to the discussion (lines 841-866; 872-886) for a more precise explanation and discussion based on the experimental results. And we corrected the spelling mistakes.

Reviewer 2 Report

Comments and Suggestions for Authors

The manuscript discusses the antioxidant activity of wheat extracts in vitro/vivo. Th topic of the work should be of interest to the journal’s readership. However, the manuscript can still be improved by inclusion of additional information in some sections.

I have the following comments to the authors:

1.     In section 3.1, line 323, add a brief explanation on why processing decreased the total phenolic content in flour and pasta.

2.     Referring to line 339, briefly add explanation to the main text on why antioxidants might be more bioavailable in certain flours during the milling process.

This sentence also requires to be rewritten. Do authors intend to refer to enhanced bioavailability “during” milling or “pursuant to /after” milling?

3.     In line 97, the abbreviation “LPS” should be explained in full term.

4.     In lines 11-115, an unnecessary hyphen exist in words “optimal”, “representativeness” and “biochemical”. It should be removed.

Author Response

The manuscript discusses the antioxidant activity of wheat extracts in vitro/vivo. Th topic of the work should be of interest to the journal’s readership. However, the manuscript can still be improved by inclusion of additional information in some sections.

I have the following comments to the authors:

  1. In section 3.1, line 323, add a brief explanation on why processing decreased the total phenolic content in flour and pasta.

The authors thank the reviewer for his appreciable comment. We have now added a detailed explanation on the effect of processing on total phenolic content, in lines 334-340.

  1. Referring to line 339, briefly add explanation to the main text on why antioxidants might be more bioavailable in certain flours during the milling process. This sentence also requires to be rewritten. Do authors intend to refer to enhanced bioavailability “during” milling or “pursuant to /after” milling?

Thank you for the comment. A detailed explanation on the bioavailability of antioxidants compounds and influencing factors on wheat-derived products has been better discussed in lines 348-359.

  1. In line 97, the abbreviation “LPS” should be explained in full term.

Thank you for the comment. The abbreviation LPS has been explained in full terms

  1. In lines 11-115, an unnecessary hyphen exist in words “optimal”, “representativeness” and “biochemical”. It should be removed.

Thank you for the comment. Unnecessary hyphen have been removed

Round 2

Reviewer 1 Report

Comments and Suggestions for Authors

The revised version of the manuscript entitled “In vitro and in vivo antioxidant and immune stimulation activity of wheat product extracts” by Beatrice Mengoni et al examined the antioxidant immune response of various parts (seed, flour, chaff, pasta) of wheat and discussed the differences of their biological activities.

According to the comments and suggestions to the reviewer’s comments, the authors revised the manuscript. Although it might be necessary to carry out some more studies to clarify the differences (different biological activities), it will be the next challenge.

The authors also pay much attention for the origin of wheats and seasonal differences of the samples. There are many extraction methods, the usage of alcohol and water is the common method and easily to apply. However, it has also some limitation, in this meaning, several other trials might be recommended, such as a use of free-dry, high-pressure, and supercritical extraction (or combination with these).

Tables

The authors added the precise explanation of the results in Table 3 (line 344-364), however, the difference of DPPH radical scavenging activity and ABTS is not still clear. It might be dependent on the radical size (molecular size of DPPH radical is not small), in this meaning it is recommendable to use galvinoxyl radical in the next time for the reference.

Figures

In figures 3-6, the difference of biological activities are observed, the results are potentially very interesting enough to discuss for the next step. As pointed out in the methods, the biological activities (activity) of each sample might be dependent on the procedure to extract. The authors also added this meaning in the reply to the reviewer letter. “These results are congruent with TPC and TFC values obtained by spectrophotometry for pasta, suggesting that the loss of phenol-rich bran fraction during the initial milling process could explain the difference with the raw matrices.” So, the authors should be very careful to handle with the samples. Similar differences are also observed in the case of flour in IL-6 and IL-10. The reviewer also wants authors to pay attention for these findings and make clear description in the next manuscript.

In Figure 7 and 8, the authors did not show clear explanation for the indicated point by the reviewer. In vivo system, many physiological reactions are taking place, it is difficult to specify the relation of the amount of biological materials and physiological reaction. However, it is also important to keep in mind to consider the physiological reaction in the molecular level.

In Figure 10, the results are quite interesting to discuss, the authors quoted some references and try to make the explanation. The logic is in a sense reasonable, however, as the author wrote in the reply “However, although its content is similar to that of seeds, we cannot exclude the presence of trace compounds not detectable by NMR that may still exhibit immunostimulatory activity”. Further studies might be necessary to enhance the level of the manuscript in the future.

Author Response

REVIEWER 1

The manuscript entitled “In vitro and in vivo antioxidant and immune stimulation activity of wheat product extracts” by Beatrice Mengoni et al examined the antioxidant immune response of various parts (seed, flour, chaff, pasta) of wheat and discussed the differences of their biological activities.

They used the hydroalcoholic extraction method (ethanol/water 80:20), from the chemical standpoint of view, there are many methods to extract biological substances from the plant materials, in this meaning, it is of interest to use the different the extraction method in the future.

The authors carried out many experiments and it is quite important to find the real active biological substrate and the part. However, the biological activities are different in many cases, it is quite difficult to focus on the one part. Some of these come from the many different types of experiments and put all these experimental results together.

They carried out many experiments, however, the discussion part in the manuscript is not so long. The authors quoted other references, which is quite important, however, they should discuss the relation of each experimental result. For example, the antioxidant activity and immune response, antioxidant activity and the amount of flavonoid content, there are many things to discuss based on their original experimental results.

Tables 

In Table 3, The authors discussed the radical scavenging activity and TPC and TFC content. The figures used in the manuscript are different from the figures of Table 3. (lines 337-345) It is quite important point to clarify. In comparison of the total amount of TPC and TFC of seed and chaff, the values are larger chaff than seeds. However, the DPPH radical scavenging activity is not in this order. Some other factors might be responsible for this, it is desirable to respond this point. 

The authors greatly thank the reviewer for his appreciable comments, and apologize for possible misunderstandings generated by the non-correspondence of the results entered in the table with those discussed in the manuscript. The authors have now completely revised this part, providing the correct information. In particular, in lines 334-340, discussions regarding the influence of processing on total polyphenol content have been expanded. In addition, the results regarding the antioxidant fraction have been completely revised and rewritten in lines 345-365, placing special emphasis on the factors that can affect the antioxidant activity of wheat products compared to raw materials.

In Table 5, the authors reported many biological samples detected by NMR spectroscopy and quantized the amount of each substrates. This information is quite important to discuss in the future project. For example, many compounds are only found in chaff. (caffeic acid, gallic acid, 4-hydroxybenzoic acid and trimethiamine). In the case of amino acids, chaff showed the highest amino acid amount in many cases, while flour showed the lower amount in almost all cases. Similar experimental results are also found in organic acids. If possible, the authors give some explanation for these points. 

Chaff, seed and flour are different entities from a botanical point of view. Indeed, chaff is the tegument of the seed, therefore it is enriched in phenols with a potential antimicrobial activity (i.e. gallic acid and 4-hydroxybenzoic acid) while there is a rather small water content. For seed, we consider the portioned by the germ and the endosperm, rich in terms of proteins (not recovered by the ethanolic extract) and other nutrients for the embryo. Flour is processed only from the endosperm and not the whole grain, therefore a difference in the content of both amino and organic acids is to be expected.

Moreover, pasta is derived from an industrial process in the presence of water that could increase some enzymatic reactions and consequently the number of free amino acids, the only ones to be detected by the method used

Figures

In Figure 3, the authors examined the inflammatory response of seeds, flour and pasta with or without administration of LPS, the trend of seeds and flour showed a similar behavior but pasta showed the low (no) inhibitory activity toward LPS administration. It is not obvious of the inflammatory response and TPC and TFC values (Table 3) and the compounds responsible for these responses. All these in vitro works are found in Fig. 4-6.

The authors would like to thank the reviewer for this comment which provided the opportunity to discuss and correlate the results of the work more comprehensively. Figure 3 shows the mRNA expression of pro- and anti-inflammatory cytokines. Pasta at 100ng/ml  reduces by 50% the IL-1beta mRNA expression, whereas at 10ng/ml the difference is not significant, probably for the high standard deviation Pasta  is effective in reducing TNF-alpha mRNA expression, in particular, a 40% reduction was observed with a concentration of 100ng/ml and a 90% reduction with a concentration of 10ng/ml. In inhibiting the expression of IL-6 mRNA, pasta behaves similarly to seeds and flour. This anti-inflammatory activity agrees the content of TPC and TFC, respectively. In addition all samples induce the expression of Arg-1 which is a marker of M2 polarization of innate immunity cells and down-regulate the expression of iNOS, a marker of M1 polarized innate immunity cells. These results are congruent with TPC (Total Complex Phenols) and TFC (Total Complex Flavonoids) values obtained by spectrophotometry for pasta, suggesting that the loss of phenol-rich bran fractions during the initial milling process could explain the difference with the raw matrices.

We believe that our data underline the possibility that bioactive compounds derived from Senatore Cappelli wheat and its processed products can counteract inflammatory reactions, especially chronic low-grade ones that insidiously increase the risk of developing chronic-degenerative diseases

Following the reviewer's notes, whom we thank, an entire paragraph focused on this issue has been added (lines 841-866)

In Figure 7 and 8, the authors examined Ros level and antioxidant enzyme expression of C. elegans, all these results showed moderate antioxidant defense system was induced by the addition of the extracts. The difference of the extracts are not corresponding with the of TPC and TFC values reported in Table 3.

We thank the Reviewer and we agree with the comment. While all the extracts demonstrated antioxidant activity in C. elegans under induced stress conditions, the lack of correspondence between the TPC and TFC values and the in vivo response may be due to the presence of other bioactive components in the extracts that can influence the antioxidant response. These components, potentially acting synergistically or through mechanisms independent of TPC and TFC, could modulate the in vivo effects.

In Figure 10, the authors showed the interesting results in the case of daf-16 mRNA expression, namely, the administration of pasta extract increase the mRNA level 50%, while the administration of seed extract reduced the mRNA expression 80%. This difference is important to discuss, although the authors briefly write as follow “,which could indicate a suppression of the IIS-mediated stress response, maybe activation other stress responses”(line 563-565). In the case of sek-1, chaff extract showed the highest increase, and in the case of pmk-1 flour showed the highest increase. These results demonstrate that the different compound is responsible for each response. It is quite meaningful to discuss these points with the compound list on Table 5. 

We thank the Reviewer for the comment and agree that the observed differences in daf-16, sek-1, and pmk-1 expression are significant and need a deeper discussion. The manuscript has been modified as follows " Regarding daf-16, the contrasting effects of pasta and seed extracts could be attributed to differences in their phytochemical profiles. As shown in Table 5, pasta extracts contain higher levels of malic acid, which may enhance the IIS-mediated stress response by promoting daf-16 activity 10.1371/journal.pone.0058345. In contrast, the high levels of amino acids such as tryptophane in seed extracts might suppress IIS signaling and redirect the stress response towards alternative pathways, since tryptophan-mediated beneficial effects in C.elegans  was independent of DAF-16/FOXO and SKN-1/Nrf2 signaling 10.1186/s12863-015-0167-2. For sek-1, the pronounced increase with chaff extract might be linked to its higher content of phenolic acids, such as caffeic and gallic acids, as these are known to activate stress response pathways like p38 MAPK  10.1007/s10522-011-9334-7, 10.1016/j.fochx.2024.101233. Similarly, the higher pmk-1 expression observed with flour extract could be due to its relatively balanced composition of antioxidants, which may synergistically enhance this pathway. However, although its content is similar to that of seeds, we cannot exclude the presence of trace compounds not detectable by NMR that may still exhibit immunostimulatory activity.”

The authors carried out many experiments and it is quite important for further studies, however, the experimental results well not well-organized enough. More precise explanation and discussion based on the experimental results are required in the revised version. There are some primitive spelling mistakes, which are easily revised by the use of word checking system. In general, it is not difficult to read the manuscript (quality is (very) good).

We really thank the reviewer for his comments We included two new paragraphs to the discussion (lines 841-866; 872-886) for a more precise explanation and discussion based on the experimental results. And we corrected the spelling mistakes.

Reviewer 2 Report

Comments and Suggestions for Authors

My comments to the authors are addressed and I have no further comment.

Author Response

The manuscript discusses the antioxidant activity of wheat extracts in vitro/vivo. Th topic of the work should
be of interest to the journal’s readership. However, the manuscript can still be improved by inclusion of
additional information in some sections.

I have the following comments to the authors:

1. In section 3.1, line 323, add a brief explanation on why processing decreased the total phenolic
content in flour and pasta.

The authors thank the reviewer for his appreciable comment. We have now added a detailed explanation on the effect of processing on total phenolic content, in lines 322-353.

2.     Referring to line 339, briefly add explanation to the main text on why antioxidants might be
more bioavailable in certain flours during the milling process.
Thank you for the comment. A detailed explanation on the bioavailability of antioxidants compounds and influencing factors on wheat-derived products has been better discussed in lines 358-378.

This sentence also requires to be rewritten. Do authors intend to refer to enhanced bioavailability
“during” milling or “pursuant to /after” milling?

As detailed in the added paragraph (lines 339-353) the enhanced bioavailability was observed during and after milling"

2. In line 97, the abbreviation “LPS” should be explained in full term.
Thank you for the comment. The abbreviation LPS has been explained in full terms

3. In lines 11-115, an unnecessary hyphen exist in words “optimal”, “representativeness” and
“biochemical”. It should be removed.
Thank you for the comment. Unnecessary hyphen were removed